# A tRNA modification balances carbon and nitrogen metabolism by regulating phosphate homeostasis

Ritu Gupta[1], Adhish S Walvekar[1], Shun Liang[2], Zeenat Rashida[1,3], Premal Shah[2]*, Sunil Laxman[1]*

[1]Institute for Stem Cell Science and Regenerative Medicine (inStem), Bangalore, India; [2]Department of Genetics, Rutgers University, Piscataway, United States; [3]Manipal Academy of Higher Education, Manipal, India

**Abstract** Cells must appropriately sense and integrate multiple metabolic resources to commit to proliferation. Here, we report that *S. cerevisiae* cells regulate carbon and nitrogen metabolic homeostasis through tRNA $U_{34}$-thiolation. Despite amino acid sufficiency, tRNA-thiolation deficient cells appear amino acid starved. In these cells, carbon flux towards nucleotide synthesis decreases, and trehalose synthesis increases, resulting in a starvation-like metabolic signature. Thiolation mutants have only minor translation defects. However, in these cells phosphate homeostasis genes are strongly down-regulated, resulting in an effectively phosphate-limited state. Reduced phosphate enforces a metabolic switch, where glucose-6-phosphate is routed towards storage carbohydrates. Notably, trehalose synthesis, which releases phosphate and thereby restores phosphate availability, is central to this metabolic rewiring. Thus, cells use thiolated tRNAs to perceive amino acid sufficiency, balance carbon and amino acid metabolic flux and grow optimally, by controlling phosphate availability. These results further biochemically explain how phosphate availability determines a switch to a 'starvation-state'.

DOI: https://doi.org/10.7554/eLife.44795.001

*For correspondence:
premal.shah@rutgers.edu (PS);
sunil@instem.res.in (SL)

Competing interests: The authors declare that no competing interests exist.

## Introduction

Cells utilize multiple mechanisms to sense available nutrients, and appropriately alter their internal metabolic state. Such nutrient-sensing systems assess internal resources, relay this information to interconnected biochemical networks, and control global responses that collectively reset the metabolic state of the cell, thereby determining eventual cell fate outcomes (*Jeong et al., 2000*; *Förster et al., 2003*; *Zaman et al., 2008*; *Broach, 2012*; *Cai and Tu, 2012*; *Ljungdahl and Daignan-Fornier, 2012*). However, much remains unknown about how cells sense and integrate information from multiple nutrient inputs, to coordinately regulate the metabolic state of the cell and commit to different fates.

In this context, the metabolic state of the cell is also closely coupled with mRNA translation. Protein synthesis is enormously energy consuming, and therefore must be carefully regulated in tune with nutrient availability (*Warner, 2001*). Generally, overall translational capacity and output increases during growth and proliferation (*Jorgensen et al., 2004*), and decreases during nutrient limitation (*Wullschleger et al., 2006*). Signalling processes that regulate translational outputs (such as the TORC1 and PKA pathways) are well studied (*Wullschleger et al., 2006*; *Zaman et al., 2008*; *Broach, 2012*; *González and Hall, 2017*). Notwithstanding this, little is known about how core components of the translation machinery might directly control metabolic outputs, and thus couple metabolic states with physiological cellular outcomes.

**eLife digest** The building blocks of all cells are made from a handful of chemical elements, including carbon, nitrogen, sulfur and phosphorus. To grow optimally, cells need to regulate their metabolism – in other words, the biochemical reactions that keep them alive – based on the availability of these elements. As a result, cells have evolved various mechanisms to sense when usable forms of these elements are present.

Proteins are chains of building blocks known as amino acids, which are assembled with the help of molecules called transfer ribonucleic acids, or tRNAs for short. Some of these molecules can be modified by attaching sulfur-containing chemical tags known as thiol groups to make "thiolated tRNAs". Research has shown that, when there was more of an amino acid known as methionine around, the cells made more thiolated tRNA. These previous studies also suggested that mutant cells lacking thiolated tRNAs might have altered carbon and nitrogen metabolism. Yet, it remained unclear what exactly was leading to this metabolic rewiring.

Now, Gupta et al. have combined several biochemical and genetics approaches to study the role of thiolated tRNAs in yeast. The experiments revealed that mutant cells lacking thiolated tRNAs were unable to properly sense the levels of methionine and other amino acids, which are the cell's major source of nitrogen. These mutant cells were also found to have a reduced level of phosphorous-containing compounds known as phosphates, which are involved in numerous biological processes.

Gupta et al. showed that reducing the level of phosphates caused carbon that is normally used to make chemicals required for growth to be re-routed towards making carbohydrates to store energy instead. This is similar to what happens when the cells are starving, showing that a 'squeeze' on internal phosphates metabolically rewires cells into a state that is like starvation.

These findings show how modified tRNAs can use the availability of amino acids to alter the cell's metabolism by altering how much phosphate is present. In doing so, the thiolated tRNAs essentially allow the cell to decide whether it has enough of the right nutrients to grow. These findings may also have implications for human health, since errors in coordinating metabolism are responsible for certain medical conditions including several cancers.

Finally, technical challenges mean many questions remain unanswered about how phosphate levels are regulated within cells. These new findings point to a pressing need to understand phosphate metabolism as a prerequisite to better understand how cells regulate their overall metabolism.

DOI: https://doi.org/10.7554/eLife.44795.002

tRNAs are core components of the translation machinery, and are extensively modified post-transcriptionally (*Björk et al., 1987*; *Phizicky and Hopper, 2010*). Some tRNA modifications are required for tRNA folding, stability, or the accuracy and efficiency of translation (*Phizicky and Hopper, 2010*). However, the roles of many of these highly conserved modifications remain unclear. One such modification is a thiolation of uridine residue present at the wobble-anticodon ($U_{34}$) position of specifically glu-, gln- and lys- tRNAs ($s^2U_{34}$) (*Gustilo et al., 2008*; *Phizicky and Hopper, 2010*). In yeast, this is mediated by a group of six enzymes- Nfs1, Tum1, Uba4, Urm1, Ncs2 and Ncs6, which are evolutionarily conserved (*Nakai et al., 2008*; *Leidel et al., 2009*; *Noma et al., 2009*). These enzymes incorporate a thiol group derived directly from an amino acid (cysteine), and replace the oxygen present at the 2-position of $U_{34}$ with sulfur (*Schmitz et al., 2008*; *Leidel et al., 2009*; *Noma et al., 2009*). Surprisingly, these thiolated tRNAs appear to have a relatively minor role in general translation, as seen in multiple studies (*Rezgui et al., 2013*; *Zinshteyn and Gilbert, 2013*; *Klassen et al., 2016*; *Chou et al., 2017*) with modest roles in enhancing the efficiency of wobble base codon-anticodon pairing (*Yarian et al., 2002*; *Rezgui et al., 2013*).

Contrastingly, tRNA thiolation appears to directly alter cellular metabolism, but this connection has remained largely unexplored. The suggestive connections to metabolism come from disparate studies. The loss of tRNA thiolation results in hypersensitivity to oxidative agents, and the TORC1 inhibitor rapamycin (*Fichtner et al., 2003*; *Goehring et al., 2003a*; *Goehring et al., 2003b*; *Laxman and Tu, 2011*; *Scheidt et al., 2014*), suggesting a role for thiolated tRNAs in maintaining

metabolic homeostasis. More pertinently, several studies have observed that a loss of this modification alters amino acid homeostasis, and the amino-acid starvation regulator Gcn4 is induced (*Laxman et al., 2013*; *Zinshteyn and Gilbert, 2013*; *Nedialkova and Leidel, 2015*). Further, thiolated tRNAs are required to maintain metabolic cycles in yeast (*Laxman et al., 2013*). Finally, the amounts of thiolated tRNAs reflect the intracellular availability of sulfur-containing amino acids (cysteine and methionine), and couple the sensing of amino acid sufficiency with growth (*Laxman et al., 2013*). These studies all hint that a core function of this tRNA modification may be to integrate the sensing of amino acid availability (primarily methionine/cysteine), with the coordinate regulation of overall metabolic state, in order for the cell to appropriately commit to growth. Yet, how thiolated tRNAs regulate metabolism, and the extent to which this may control cellular outcomes remains entirely unaddressed.

In this study, by directly analyzing different metabolic outputs, we identify the metabolic nodes that are altered in tRNA thiolation deficient cells. We find that tRNA thiolation regulates central carbon and nitrogen (amino acid) metabolic outputs, by controlling flux towards storage carbohydrates. In tRNA thiolation deficient cells, overall metabolic homeostasis is altered, with carbon flux diverted away from the pentose-phosphate pathway and nucleotide synthesis axis, and towards storage carbohydrates trehalose and glycogen. This thereby alters cellular commitments towards growth and cell cycle progression. Counter-intuitively, we discover that this metabolic-state switch in cells lacking tRNA thiolation is achieved by down-regulating a distant metabolic arm of phosphate homeostasis. We biochemically elucidate how regulating phosphate balance can couple amino acid and carbon utilization towards or away from nucleotide synthesis, and identify trehalose synthesis as a pivotal control point for this metabolic switch. Through these findings we show how tRNA thiol-modifications regulate overall metabolic homeostasis, integrating nutrient inputs to enable optimal growth. We further present a general biochemical explanation for how inorganic phosphate homeostasis regulates commitments to different arms of carbon and nitrogen metabolism, thereby determining how cells commit to a 'growth' or 'starvation' state.

## Results

### Amino acid and nucleotide metabolism are decoupled in tRNA thiolation deficient cells

Earlier studies had observed an increased expression of amino acid biosynthetic genes, and an activation of the amino acid starvation responsive transcription factor Gcn4, in cells lacking tRNA thiolation (*Laxman et al., 2013*; *Zinshteyn and Gilbert, 2013*; *Nedialkova and Leidel, 2015*). These studies therefore suggested that tRNA thiolation-deficient cells were amino-acid starved. We investigated this surmise, by directly measuring free intracellular amino acids in wild-type (WT) and tRNA thiolation mutant cells, using two distinct thiolation pathway mutants (*uba4Δ* and *ncs2Δ*). In our studies, we used a prototrophic yeast strain grown in synthetic minimal medium without supplemented amino acids, in order to minimize any confounding interpretations coming from supplemented amino acids/nucleobases in the medium. Using quantitative, targeted LC-MS/MS approaches, we compared relative amounts of amino acids in WT and thiolation mutants. Unexpectedly, we observed a substantial increase in the intracellular pools of free amino acids in thiolation mutants (*Figure 1A*). This shows that the thiolation deficient cells are not amino acid starved, but instead accumulate amino acids. We next correlated these actual amino acid amounts with the abundance of Gcn4. Gcn4 is the major amino acid starvation responsive transcription factor, and is induced upon amino acid starvation (*Hinnebusch, 1984*; *Hinnebusch, 2005*) (*Figure 1B*). We measured Gcn4 protein in WT and thiolation deficient cells, and contrarily observed increased Gcn4 protein in thiolation mutants (*Figure 1C*). Further, *GCN4* translation was correspondingly higher in thiolation mutants (*Figure 1—figure supplement 1A*) (as also seen earlier in *Zinshteyn and Gilbert, 2013*; *Nedialkova and Leidel, 2015*). This increased *GCN4* translation in the thiolation mutants was also Gcn2- and eIF2α phosphorylation-dependent (*Figure 1—figure supplement 1B and C*). These observations comparing actual amino acid amounts in cells with the activity of Gcn4 therefore present a striking paradox. As canonically understood, Gcn4 is induced upon amino acid starvation, while Gcn4 translation and protein decrease when intracellular amino acid amounts are restored (*Hinnebusch, 1984*; *Hinnebusch, 2005*). Contrastingly, in the results observed here, despite the high

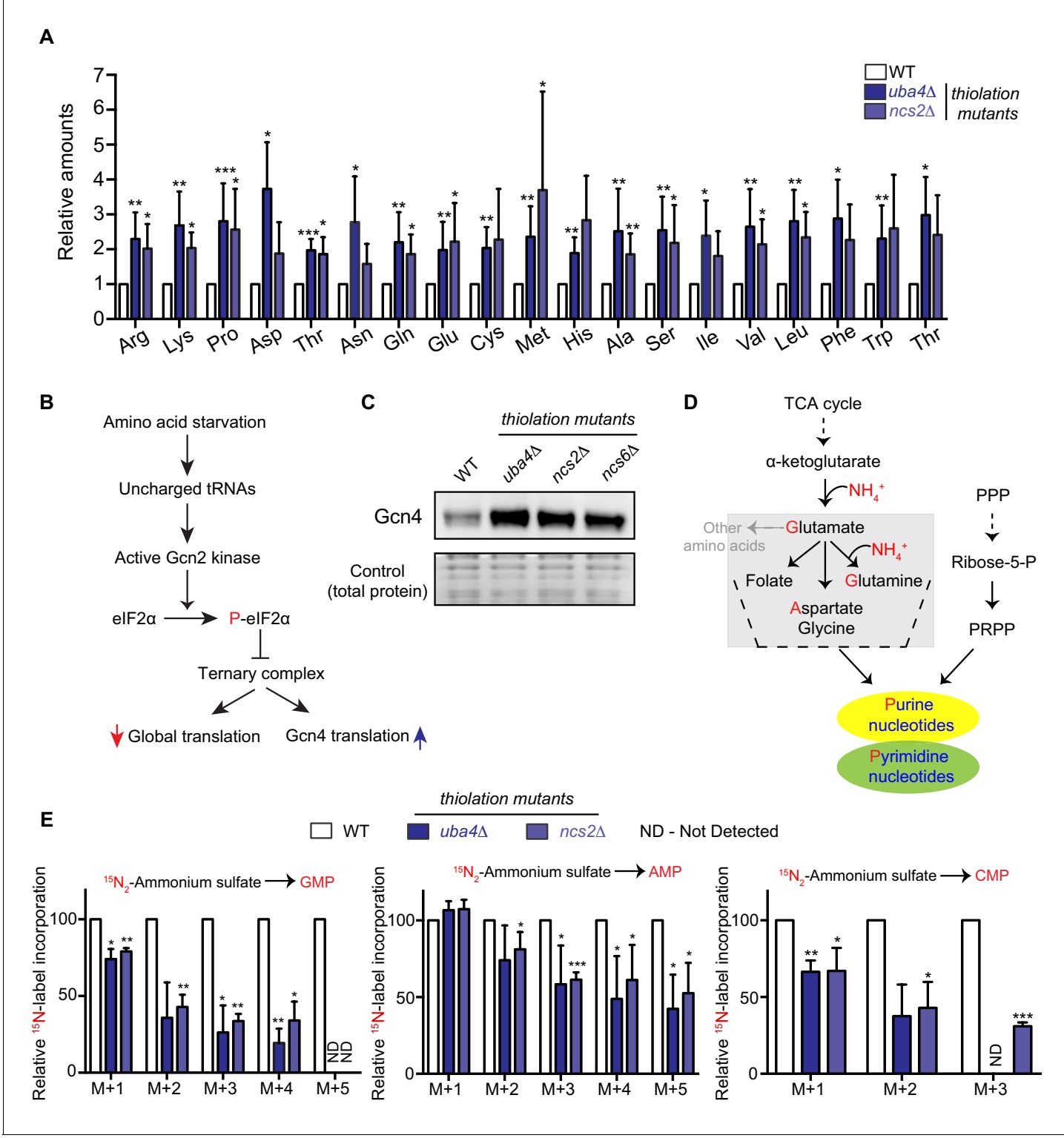

**Figure 1.** Amino acid and nucleotide metabolism are decoupled in tRNA thiolation deficient cells. (A) Intracellular pools of amino acids are increased in tRNA thiolation mutants. Steady-state amino acid amounts were measured in wild-type (WT) and tRNA thiolation mutant cells (*uba4Δ* and *ncs2Δ*) grown in minimal media by targeted liquid chromatography/mass spectrometry (LC-MS/MS). Amino acid levels in tRNA thiolation mutant cells relative to WT are plotted, where levels in WT were set to 1. * denotes statistical significance (Student's t-test), comparing tRNA thiolation mutant cells (*uba4Δ* and *ncs2Δ*) to WT. Data are displayed as mean ± SD, n >= 3. *p<0.05, **p<0.01, ***p<0.001. (B) A schematic representation illustrating the induction of Gcn4 translational upon amino acid starvation, as mediated by the Gcn2 kinase, and phosphorylation of the eIF2α initiation factor. (C) Gcn4 protein is

*Figure 1 continued on next page*

*Figure 1 continued*

increased in tRNA thiolation mutants. Western blots indicating Gcn4 protein levels (Gcn4 tagged with HA epitope at the endogenous locus) in WT and tRNA thiolation mutant cells (*uba4Δ*, *ncs2Δ* and *ncs6Δ*) grown in minimal media, as detected using an anti-HA antibody. A representative blot obtained from three biological replicates (n = 3) is shown. Also see *Figure 1—figure supplement 1*. (D) A schematic representation of de novo nucleotide (purine and pyrimidine) biosynthesis from its precursors- amino acids, the pentose phosphate pathway and PRPP (5-Phopsphoribosyl-1-Pyrophosphate), and the folate/one-carbon pathway. (E) Nucleotide synthesis is decreased in tRNA thiolation mutants (nitrogen label). WT and tRNA thiolation mutant cells (*uba4Δ* and *ncs2Δ*) grown in minimal media were pulse-labelled with $^{15}N_2$-labelled ammonium sulfate for 90 min to measure newly synthesized nucleotides (GMP, AMP and CMP) using targeted LC-MS/MS. The incorporation of $^{15}N$ atoms from $^{15}N_2$-labelled ammonium sulfate into nucleotides is represented as M + n, where n is the number of $^{15}N$-labelled atoms. Label incorporation in tRNA thiolation mutant cells relative to WT is plotted, where label incorporation in WT was set to 100. * denotes statistical significance (Student's t-test), comparing tRNA thiolation mutant cells (*uba4Δ* and *ncs2Δ*) to WT. Data are displayed as means ± SD, n = 3 for AMP and CMP, n = 2 for GMP. *p<0.05, **p<0.01, ***p<0.001. Also see *Figure 1—figure supplement 2A,B and D*.

DOI: https://doi.org/10.7554/eLife.44795.003

The following figure supplements are available for figure 1:

**Figure supplement 1.** Gcn2-dependent translational induction of Gcn4 in tRNA thiolation deficient cells.

DOI: https://doi.org/10.7554/eLife.44795.004

**Figure supplement 2.** Amino acid and nucleotide measurements in tRNA thiolation deficient cells.

DOI: https://doi.org/10.7554/eLife.44795.005

amino acid amounts present in the tRNA thiolation mutants, the Gcn2-Gcn4 pathway remains induced. We therefore concluded that the metabolic node regulated by tRNA thiolation, resulting in an apparent amino acid starvation signature, cannot be at the level of amino acid biosynthesis and availability.

We therefore considered the possible metabolic outcomes of amino acid utilization, and hypothesized that an alteration in amino acid utilization could be a source of this metabolic rewiring. In particular, amino acids are the sole nitrogen donors for de novo nucleotide synthesis (*Figure 1D*). Since amino acids accumulated in thiolation mutants, we explored the possibility that this was due to reduced de novo nucleotide synthesis. We first measured steady-state nucleotides in WT and thiolation mutants, and observed decreased steady-state levels of nucleotides (NMPs, as well as ATP) in thiolation mutants (*Figure 1—figure supplement 2A and B*). To unambiguously determine if these decreased nucleotide amounts in the thiolation mutants were due to reduced nucleotide synthesis, we adopted a metabolic flux-based approach we had developed earlier (*Walvekar et al., 2018b*). In such an approach, $^{15}N$-labelled ammonium sulfate can be provided as a pulse to cells growing with ammonium sulfate as a sole nitrogen source, and label incorporation via glutamine and aspartate into newly formed nucleotides can be measured. Notably, the incorporation of $^{15}N$-label into nucleotides (GMP, AMP and CMP) decreased in thiolation mutants relative to WT cells (*Figure 1E*), indicating reduced flux towards nucleotide synthesis. As an internal control, the $^{15}N$-label incorporation into amino acids (aspartate and glutamine) themselves were not affected in thiolation mutants (*Figure 1—figure supplement 2C*), ruling out amino acid synthesis defects. These data therefore show that nitrogen incorporation from amino acids to nucleotides decrease in the thiolation mutants, resulting in decreased nucleotides. We further addressed this pharmacologically, using a purine-analog, 8-azaadenine, which acts as a pseudo-feedback inhibitor of nucleotide biosynthesis. Consistent with the decreased nucleotide levels observed, thiolation mutants exhibited increased sensitivity to 8-azaadenine (*Figure 1—figure supplement 2D*). Collectively, these data show that the loss of tRNA thiolation decreases nucleotide biosynthesis, with a corresponding accumulation of amino acids. Notably, early studies have shown that nucleotide limitation can itself directly induce Gcn4 (*Rolfes and Hinnebusch, 1993*), suggesting that the increased Gcn4 amounts could be due to this. We also asked if the decreased nucleotide synthesis was due to reduced expression of nucleotide biosynthetic genes. We observed that the expression of candidate genes in this pathway were increased in thiolation mutants (*Figure 1—figure supplement 2E*), diminishing the possibility of reduced nucleotide biosynthetic capacity as a reason for decreased nucleotides. Indeed, increased mRNA levels of nucleotide biosynthetic genes observed in thiolation mutants may be due to feedback upregulation in response to reduced nucleotides, which is also a well-established phenomenon (*Davis and Ares, 2006*; *Kwapisz et al., 2008*).

Collectively, despite increased intracellular pools of amino acids, tRNA thiolation deficient cells exhibit signatures of amino acid starvation, including decreased nucleotide biosynthesis. These data therefore suggest that the tRNA thiolation pathway is important for cells to appropriately balance amino acid utilization for nucleotide synthesis.

## Carbon flux is routed towards storage carbohydrates in thiolation mutants

Despite amino acids being non-limiting in thiolation deficient cells, flux towards nucleotide synthesis was decreased. This observation was in itself puzzling, and the reason was not obvious. We therefore asked if carbon metabolism was instead rewired in the thiolation mutants. Our reasoning was as follows: while amino acids are the sole nitrogen donors for nucleotide synthesis, the carbon backbone for nucleotides is derived from central carbon metabolism (*Figure 2A*). We reasoned that since the decreased nucleotide synthesis was not due to amino acid limitation, this could instead be due to a metabolic shift where carbon flux is routed away from nucleotide synthesis. Carbon derived from glucose is converted to glucose-6-phosphate, and then is typically directed towards glycolysis and the pentose phosphate pathway (PPP). The PPP, along with one-carbon folate metabolism provides the necessary carbon precursors (including ribose sugars) for nucleotide synthesis (*Figure 2A*) (*Boyle, 2005*; *Hosios and Vander Heiden, 2018*). However, if glucose-6-phosphate is instead diverted towards storage carbohydrates trehalose and glycogen (*Figure 2A*), this can result in reduced flux into the arm of glucose metabolism leading towards nucleotide synthesis. To assess if such a metabolic rewiring might happen in tRNA thiolation mutants, we pulsed [U-$^{13}$C$_6$]-labelled glucose to growing WT or thiolation deficient cells, and measured label incorporation into nucleotides as the end-point readout. Here, labelled carbons will only be present in newly synthesized nucleotides, and the label can only come from the pulsed labelled glucose, through the PPP and one-carbon metabolic pathways (*Figure 2A*). We observed significantly decreased carbon label incorporation towards new nucleotide synthesis, as shown for GMP and AMP, as well as label incorporation into ADP and ATP in the thiolation mutants (*Figure 2B*, *Figure 2C*, and *Figure 2—figure supplement 1A*). This result is also consistent with the decreased nucleotide synthesis based on amino acid derived nitrogen assimilation, observed earlier (*Figure 1E*). Experimental note: Given how rapidly the pulsed carbon label saturates in glucose medium for early glycolytic and PPP intermediates (*Heerden et al., 2014*), using nucleotide synthesis as a read-out of this arm of carbon metabolism is a more reliable, quantitative indicator of carbon flux. Here, we reliably obtained nucleotide information (ensuring that label incorporation was not saturated) by pulsing cells with $^{13}$C-glucose and quenching/processing metabolites within 5 min. By this time, the $^{13}$C-glucose label incorporation into glycolytic and pentose phosphate pathway intermediates was already near-saturation, and hence differences in these metabolites could not be seen (*Figure 2—figure supplement 1B*). We therefore carried out experiments where cells were quenched and metabolites extracted within 2 min of $^{13}$C-glucose addition. Here, a significant decrease in $^{13}$C-label incorporation into newly synthesized ribose-5-phosphate (a late PPP metabolite) was observed (*Figure 2D*). This is entirely consistent with results obtained with nucleotides in *Figure 2B and C*. Summarizing, these data show that carbon (glucose) flux towards nucleotide synthesis was reduced in the thiolation mutants.

The end-point readouts of the alternative metabolic arm where glucose-6-phosphate is diverted away from the PPP are trehalose and glycogen (as shown in *Figure 2A*). To examine if this arm of metabolism is altered in the thiolation mutants, thereby resulting in the decreased carbon flux towards nucleotides (shown earlier), we first estimated the steady-state amounts of trehalose and glycogen with a biochemical assay. Here, we observed a marked increase in these metabolites in the thiolation mutants (*Figure 2E* and *Figure 2—figure supplement 1C*). Subsequently, we directly estimated flux towards trehalose synthesis. For this, we used a similar experiment as described earlier, where [U-$^{13}$C$_6$]-labelled glucose was pulsed and newly formed labelled trehalose was measured in a metabolic flux experiment, to test if this arm of the pathway is altered. Here, we observed a strong increase in the synthesis of trehalose (as measured by label incorporation into M + 6 and M + 12 mass isotopomers of trehalose) in the tRNA thiolation mutants (*Figure 2F*). Collectively, these results show that cells lacking tRNA thiolation rewire metabolic outputs towards the synthesis of storage carbohydrates, and away from nucleotide synthesis, suggesting a 'starvation-like' metabolic state. Notably, this occurs despite the absence of glucose (carbon) or amino acid (nitrogen) limitation in the thiolation mutants.

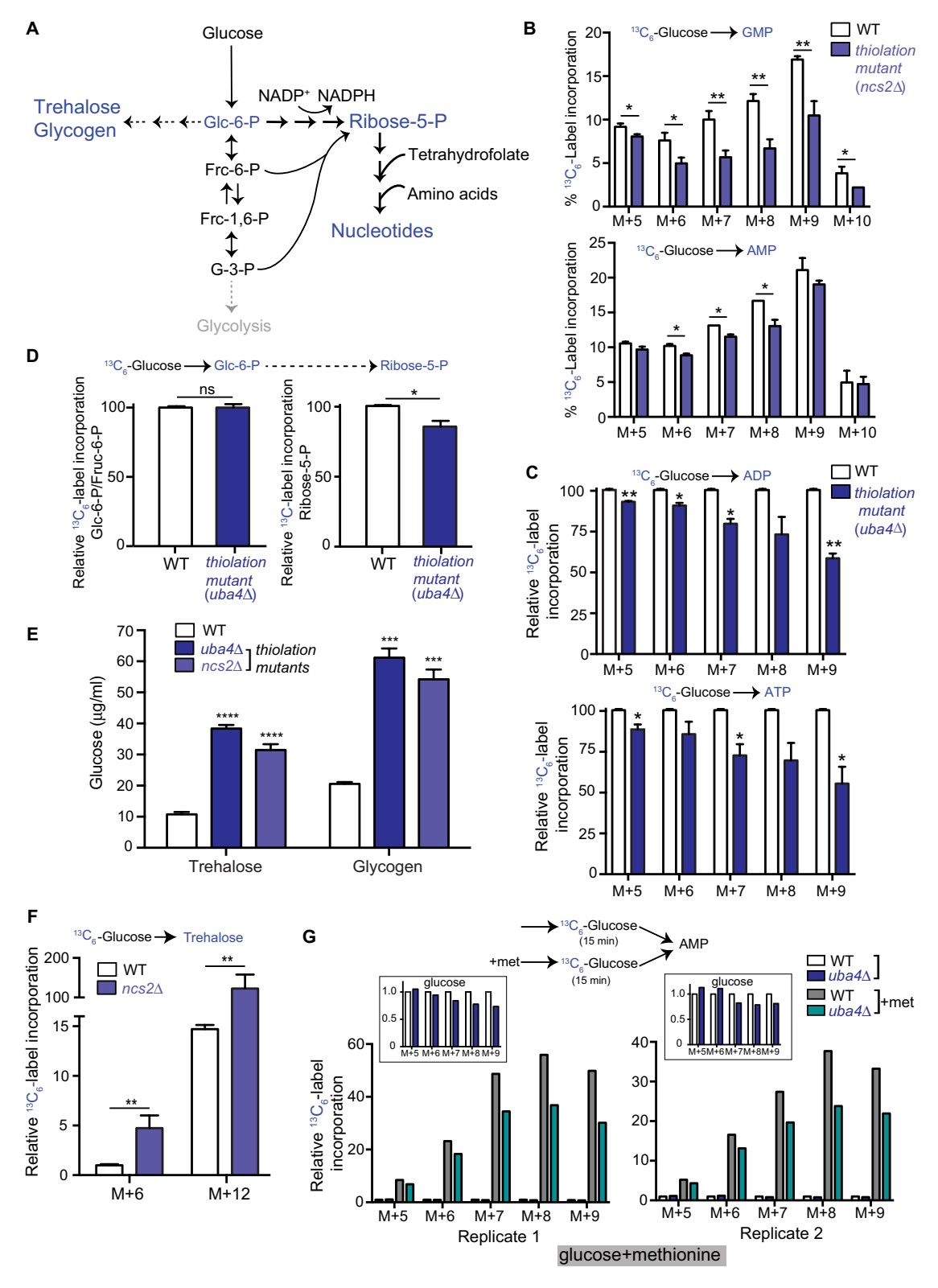

**Figure 2.** Carbon flux is routed towards storage carbohydrates in thiolation mutants. (**A**) Schematic representation depicting nucleotide and storage carbohydrates (trehalose and glycogen) biosynthesis, starting from a common precursor, glucose-6-phosphate. In normal conditions where carbon and nitrogen are not limiting, flux is typically higher towards the pentose phosphate pathway (ribose-5-phosphate), and eventually nucleotides. (**B**) Nucleotide synthesis is decreased in tRNA thiolation mutant (carbon label). WT and tRNA thiolation mutant cells (*ncs2Δ*) grown in minimal media were

*Figure 2 continued on next page*

*Figure 2 continued*

pulse-labelled with [U-$^{13}$C$_6$]-labelled glucose for 10 min, and quenched, and newly synthesized nucleotides (GMP, AMP) were measured, using LC-MS/MS. Percent label incorporation in WT and tRNA thiolation mutant cells was plotted. The incorporation of $^{13}$C atoms from [U-$^{13}$C$_6$]-labelled glucose into nucleotides is represented as M + n, where n is the number of $^{13}$C-labelled atoms (with all five carbons labelled in the ribose sugar). * denotes statistical significance (Student's t-test) of relevant data comparing tRNA thiolation mutant cells (*ncs2Δ*) to WT. Data are displayed as means ± SD, n = 3 for GMP and n = 2 for AMP. *p<0.05, **p<0.01. Also see *Figure 2—figure supplement 1A*. (C) ADP and ATP synthesis is decreased in the thiolation mutant. WT and tRNA thiolation mutant cells (*uba4Δ*) grown in minimal media were pulse-labelled with [U-$^{13}$C$_6$]-labelled glucose for 10 min, quenched, and newly synthesized ADP and ATP were measured, using LC-MS/MS. Relative label incorporation in WT and tRNA thiolation mutant cells are shown, where label amounts in WT cells is set to 1. The incorporation of $^{13}$C atoms from [U-$^{13}$C$_6$]-labelled glucose into nucleotides is represented as M + n, where n is the number of $^{13}$C-labelled atoms (with all five carbons labelled in the ribose sugar). * denotes statistical significance (Student's t-test) of relevant data comparing tRNA thiolation mutant cells (*ncs2Δ*) to WT. Data are displayed as means ± SD, n = 3. *p<0.05, **p<0.01. (D) PPP flux is decreased in the tRNA thiolation mutant. WT and tRNA thiolation mutant cells (*uba4Δ*) grown in minimal media were pulse-labelled with [U-$^{13}$C$_6$]-labelled glucose for 2 min, quenched and collected, and early (glucose-6-phosphate) and late (ribose-5-phosphate) PPP intermediates were measured by LC-MS/MS. Relative $^{13}$C$_6$-label incorporation into these metabolites are shown. Data are displayed as means ± SD, n = 2. *p<0.05, **p<0.01, Student's t-test. Also see *Figure 2—figure supplement 1B*. (E) Steady-state trehalose and glycogen amounts are increased in tRNA thiolation mutants. Trehalose and glycogen content of WT and tRNA thiolation mutant cells (*uba4Δ* and *ncs2Δ*) grown in minimal media was plotted. * denotes statistical significance (Student's t-test), comparing tRNA thiolation mutant cells (*uba4Δ* and *ncs2Δ*) to WT. Data are displayed as means ± SD, n = 4 biological replicates with two technical replicates each for trehalose, and n = 3 biological replicates with two technical replicates each for glycogen. ***p<0.001, ****p<0.0001. Also see *Figure 2—figure supplement 1C*. (F) Trehalose synthesis is increased in tRNA thiolation mutant. WT and tRNA thiolation mutant cells (*ncs2Δ*) grown in minimal media were pulse-labelled with [U-$^{13}$C$_6$]-labelled glucose for 5 min, quenched, and label incorporation into newly synthesized trehalose was measured using targeted LC-MS/MS. Relative label incorporation into trehalose in WT and tRNA thiolation mutant cells is shown, where label in WT M + 6 is set to 1. The incorporation of $^{13}$C atoms from [U-$^{13}$C$_6$]-labelled glucose into trehalose is represented as M + n, where n is the number of $^{13}$C-labelled atoms. * denotes statistical significance (Student's t-test), comparing tRNA thiolation mutant cells (*ncs2Δ*) to WT. Data are displayed as means ± SD, n = 3. *p<0.05, **p<0.01. (G) Thiolation mutants exhibit reduced methionine-induced carbon incorporation into nucleotides. WT and tRNA thiolation mutant cells (*uba4Δ*) were grown in standard minimal media, 2% glucose, with or without 2 mM methionine (met) supplemented. Cells were pulse-labelled with [U-$^{13}$C$_6$]-labelled glucose for 15 min, and metabolites extracted. Here, carbon incorporation into newly synthesized nucleotides (AMP) was measured, using LC-MS/MS. The relative label incorporation in WT and tRNA thiolation mutant cells with or without methionine is shown, where label in WT in the absence of methionine is set to 1. Two independent biological replicates (replicate 1 and 2) are shown. The relative label incorporation in WT and tRNA thiolation mutant cells growing in standard minimum medium (without methionine supplementation) is shown as an inset, to suitably illustrate the strong induction in nucleotide synthesis due to methionine supplementation. The incorporation of $^{13}$C atoms from [U-$^{13}$C$_6$]-labelled glucose into nucleotides is represented as M + n, where n is the number of $^{13}$C-labelled atoms (with all five carbons labelled in the ribose sugar). Also see *Figure 2—figure supplement 2D*.

DOI: https://doi.org/10.7554/eLife.44795.006

The following figure supplements are available for figure 2:

**Figure supplement 1.** Carbon flux is routed away from nucleotides and towards storage carbohydrates in tRNA thiolation deficient cells.

DOI: https://doi.org/10.7554/eLife.44795.007

**Figure supplement 2.** Sulfur starvation phenocopies the metabolic state of thiolation mutants.

DOI: https://doi.org/10.7554/eLife.44795.008

## Sulfur starvation phenocopies the metabolic state of thiolation mutants

Earlier studies have noted a coupling of tRNA thiolation with methionine/sulfur amino acid availability (*Laxman et al., 2013*). Here, the amount of thiolated tRNAs increase with increasing sulfur amino acids and vice versa. Further, in thiolation mutants, proteins involved in methionine salvage and biosynthesis increase even when methionine was in excess, suggesting a mis-sensing of this amino acid in these mutants. Finally, Uba4 is itself regulated by methionine amounts, decreasing sharply with slight methionine-limitation (*Laxman et al., 2013*). Separately, studies (*Tu et al., 2005*; *Laxman et al., 2013*) show that the amounts of thiolated tRNAs are highest in cells entering a 'growth state' (with high ribosomal biosynthesis). In this context, we also recently defined a methionine induced anabolic program, where abundant methionine triggers increased carbon flux (particularly PPP flux) towards nucleotide synthesis (*Walvekar et al., 2018b*). Given our observations here in standard medium, where thiolation mutants showed decreased carbon flux from glucose towards nucleotides, we further investigated this coupling of tRNA thiolation with methionine availability. The prediction is that if tRNA thiolation enables cells to fully respond to abundant methionine, then thiolation mutants will exhibit a reduced methionine response. As a result, when methionine is supplemented, thiolation mutants will show reduced carbon incorporation into nucleotides (i.e. nucleotide synthesis), compared to WT cells. To test this, WT or thiolation mutant (*uba4Δ*) cells grown in standard glucose minimal medium were supplemented with 2 mM methionine, and then pulsed with

$^{13}$C-glucose (as described earlier) for 15 min. Subsequently, we compared carbon-label incorporation into newly synthesized nucleotides. Here, we expectedly observed a sharp increase in carbon-label incorporation into nucleotides in WT cells supplemented with methionine (*Figure 2G*, and *Figure 2— figure supplement 1D*). Notably, while the extent of this methionine-dependent induction of carbon flux into nucleotides was significantly reduced, it was not completely abolished in the thiolation mutant (*uba4Δ*), indicating the involvement of additional as yet unknown pathways (*Figure 2G*, and *Figure 2—figure supplement 1D*). Together, these data reiterate that tRNA thiolation allows cells to fully integrate amino acid (methionine) sensing, with a routing of carbon (glucose) flux towards new nucleotide synthesis, collectively indicative of a growth state. Furthermore, in a converse experiment, we subjected WT cells to brief inorganic sulfur/sulfur amino acid limitation, to determine the metabolic signature of cells in this condition (experimental design shown in *Figure 2—figure supplement 2A*). Cells subjected to a brief shift to sulfur starved medium showed a sharp decrease in sulfur amino acid-related metabolites (*Figure 2—figure supplement 2B*). We measured the steady-state amounts of other amino acids, nucleotides (AMP), trehalose and the level of Gcn4 translation. We observed increased steady-state amino acid amounts (*Figure 2—figure supplement 2C*), reduced nucleotides (*Figure 2—figure supplement 2D*), increased trehalose levels (*Figure 2—figure supplement 2E*), and strong Gcn4 induction (*Figure 2—figure supplement 2F*) upon sulfur limitation. These data strikingly resembled the metabolic state of the thiolation mutants. Thus, WT cells subject to sulfur amino acid limitation phenocopied the metabolic signature of tRNA thiolation mutants.

## tRNA thiolation couples cellular metabolic state with normal cell cycle progression

Dissecting physiological roles of such a fine-tuning of metabolic outputs can be challenging, and this has been the case for tRNA thiolation mutants. However, a simple yeast system, termed 'yeast metabolic cycles' or metabolic oscillations, has been effective in identifying regulators that couple metabolism with cell growth/cell-division (*Tu et al., 2005*; *Slavov and Botstein, 2011*). In continuous, glucose-limited cultures, yeast cells exhibit robust metabolic oscillations, which are tightly coupled to the cell division cycle, and where DNA replication and cell division are restricted to a single temporal phase (*Tu et al., 2005*; *Chen et al., 2007*). In an earlier study we had observed that tRNA thiolation mutants exhibit abnormal metabolic cycles (*Laxman et al., 2013*). This was reminiscent of phenotypes exhibited by mutants of cell division cycle regulators (*Chen et al., 2007*). We therefore asked if tRNA thiolation coupled metabolic and cell division cycles. To test this, we sampled the cells at regular intervals of time during the metabolic cycles, and determined their budding index. While WT cells showed synchronized cell cycle progression, tRNA thiolation mutants showed asynchronous cell division (*Figure 3A*), suggesting a de-coupling of metabolic and cell division cycles. Given our earlier data showing a metabolic rewiring away from nucleotide synthesis in thiolation mutants, we hypothesized that tRNA thiolation controlled normal cell cycle progression by regulating the balance between nucleotide synthesis, and storage carbohydrate synthesis.

To test this directly, we arrested cells in G1-phase using alpha factor, synchronously released them into the cell cycle by washing away the alpha factor, and monitored cell cycle progression by flow cytometry (*Figure 3B*). 30 min post-release from G1-arrest, we observed delayed cell cycle progression and accumulation of cells in the S-phase in tRNA thiolation deficient cells. Further, using time lapse live-cell microscopy we found that the duration of the S-G2/M phase was longer in thiolation mutants (*Figure 3C* and *Figure 3D*). Since nucleotides are required for DNA replication during the S-phase of the cell cycle, we reasoned that that this S-phase delay is due to decreased flux towards nucleotide synthesis in thiolation mutants (shown earlier). To investigate this, we examined the sensitivity of WT and thiolation deficient cells to hydroxyurea (HU), which inhibits the ribonucleotide reductase (RNR) enzyme and arrests cells in the S-phase. Thiolation mutants exhibited increased sensitivity to HU (*Figure 3E* and *Figure 3—figure supplement 1A*). As a control, to rule out any abnormal activation of DNA damage readouts in the thiolation mutants as a cause for this phenotype, we also examined activity of the Rad53 checkpoint pathway. Indeed, the observed HU sensitivity was not due to any defect in the activation of the Rad53 checkpoint pathway in thiolation mutants (*Figure 3—figure supplement 1B*).

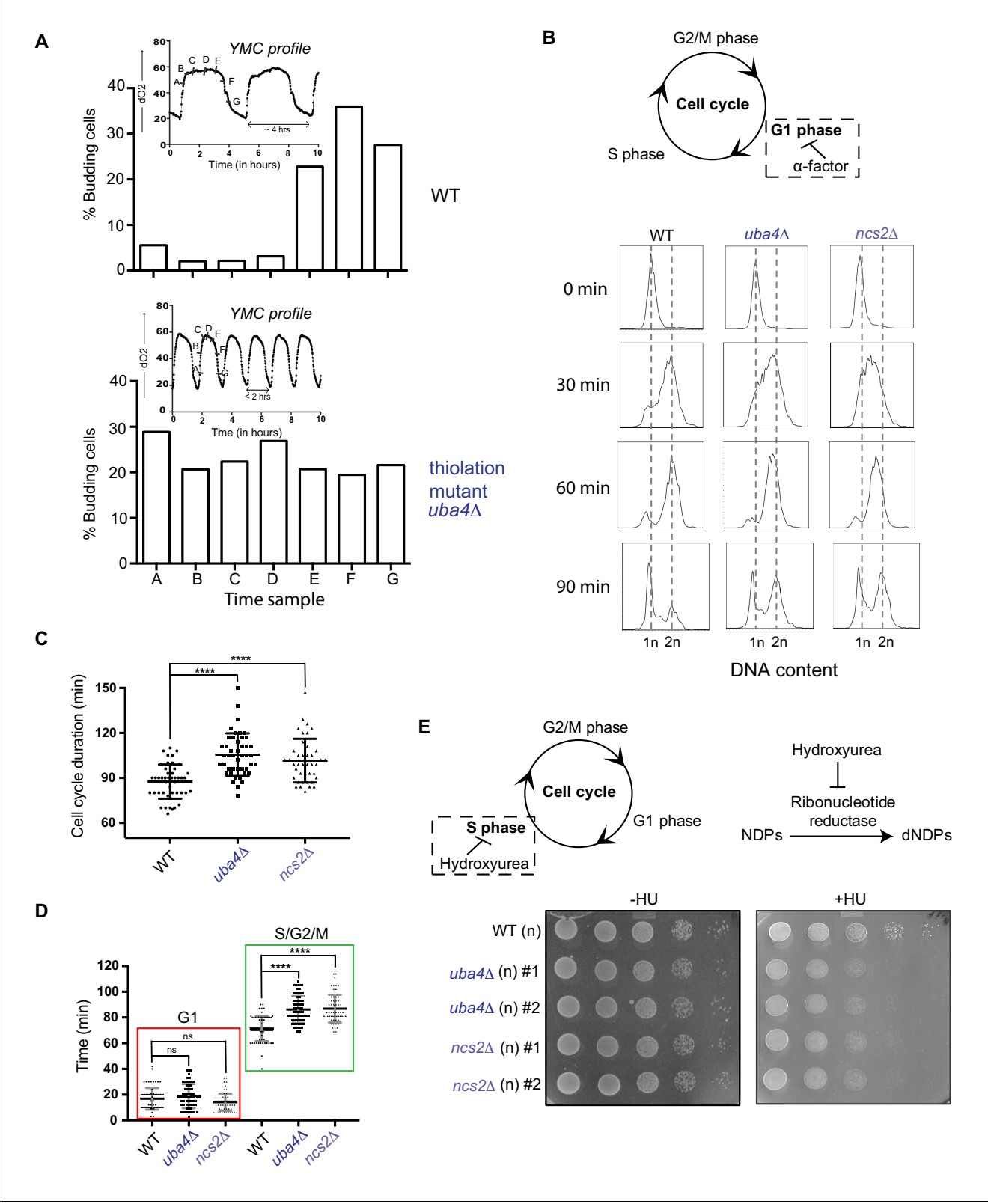

**Figure 3.** tRNA thiolation couples cellular metabolic state with normal cell cycle progression. (**A**) tRNA thiolation mutants exhibit asynchronous cell division and disrupted yeast metabolic cycles. WT and tRNA thiolation mutant cells (*uba4Δ*) growing in chemostat cultures under conditions of normal yeast metabolic cycles, were sampled at 20 min time intervals. Percentage of budded cells represents the fraction of cells in S/G2/M phases of the cell cycle. At least 200 cells were analysed for each time point, for the respective strains. Inset: the oxygen consumption profiles of WT and tRNA thiolation

*Figure 3 continued on next page*

*Figure 3 continued*

mutant metabolic cycles are represented. Note that the metabolic cycles of thiolation mutants are disrupted. (B) tRNA thiolation mutants show delayed cell cycle progression. The DNA content of WT and tRNA thiolation mutant cells (*uba4Δ* and *ncs2Δ*) grown in minimal media, during G1 arrest (0 min) and after release from G1 arrest (30, 60 and 90 min) was determined by flow cytometry analysis, and is presented. (C and D) Cell cycle duration, distribution of G1 phase duration (red) and S/G2/M phase durations (green) for WT and tRNA thiolation mutant cells (*uba4Δ* and *ncs2Δ*) were measured in single cells using time lapse live-cell microscopy. The G1 duration (starting from the time of complete division of mother and daughter cells to bud emergence), and S/G2/M durations (from the time of bud emergence to complete division of mother and daughter cells) were determined for only first generation mother cells. At least 100 cells were analysed for each strain. * denotes statistical significance (Student's t-test), comparing tRNA thiolation mutant cells (*uba4Δ* and *ncs2Δ*) to WT. ns denotes non-significant difference. ****p<0.0001. (E) tRNA thiolation mutants exhibit increased HU sensitivity. WT and tRNA thiolation mutant cells (*uba4Δ* and *ncs2Δ*) grown in minimal media were spotted on minimal media agar plates containing 150 mM HU. Also see *Figure 3—figure supplement 1A and B*.

DOI: https://doi.org/10.7554/eLife.44795.009

The following figure supplement is available for figure 3:

**Figure supplement 1.** HU sensitivity in tRNA thiolation deficient cells is independent of the Rad53 checkpoint pathway.
DOI: https://doi.org/10.7554/eLife.44795.010

---

Summarizing, these results show that tRNA thiolation-mediated regulation of metabolic homeostasis, leading towards regulated nucleotide synthesis, is required for appropriately coupling metabolic state with normal cell cycle progression.

## Loss of tRNA thiolation results in reduced phosphate homeostasis (*PHO*)-related transcripts and ribosome-footprints

Thus far, it remains unclear why the loss of tRNA thiolation results in this distinct metabolic switch, where carbon and amino acid flux is diverted away from nucleotide synthesis and into storage carbohydrates. This therefore suggests a deeper, non-intuitive regulatory check-point underpinning the overall metabolic rewiring towards a 'starvation-like' state in tRNA thiolation mutants. In order to identify what this controlling bottleneck might be, we identified transcriptional and translational changes in thiolation mutants relative to WT by performing RNA-seq and Ribo-seq (*Ingolia et al., 2009*) based on methods described earlier (*Weinberg et al., 2016*; *McGlincy and Ingolia, 2017*). Notably, as detailed in the Materials and methods section, while collecting cells for RNA and Ribo-seq, we avoided the use of cycloheximide, as described earlier (*Weinberg et al., 2016*), to minimize biases in our interpretations of ribosome profiling datasets due to the use of this translational inhibitor as described elsewhere (*Hussmann et al., 2015*). Instead, cells were rapidly harvested by filtration and lysed as described. Ribosome profiling datasets were generated for WT cells as well as two distinct thiolation pathway mutants (*uba4Δ* and *ncs2Δ*), in biological triplicates. The correlations among the biological replicates of ribosome footprint reads, and also among RNA-seq reads were all excellent (R > 0.97), as shown in *Figure 4—figure supplement 1A*. Further, figures (*Figure 4—figure supplement 1B*; *Figure 4—figure supplement 1C*) show transcript and ribosome footprint read correlations ($R^2$ in all comparisons > 0.8), as well as read-length distributions.

Using these datasets, we compared global gene expression, as well as ribosome footprints of WT cells with the *uba4Δ* and *ncs2Δ* thiolation mutants (*Figure 4A*). Notably, comparing WT cells with the thiolation mutants (*uba4Δ* and *ncs2Δ*), we surprisingly found exceptional correlation for transcripts, as well as ribosome footprints ($R^2 > 0.97$, and $p <= 2.2 \times 10^{-16}$ for all datasets) (*Figure 4A and B*). These data surprisingly revealed that there are very little gene expression or translation changes observed in the thiolation mutants (complete data in *Supplementary file 4*). Fewer than ~30 genes were up or downregulated at a two-fold change cutoff (arbitrarily used to illustrate the point), compared to WT cells (*Figure 4A and B*), with a false discovery rate (FDR) of 0.05. Furthermore, we observe only modest increases in ribosome-densities at codons recognized by thiolated tRNAs – AAA, CAA and GAA in the *uba4Δ* and *ncs2Δ* cells (*Figure 4—figure supplement 2A*). Collectively, these extensive analysis show that the loss of tRNA thiolation has minimal effects on translational outputs in vivo, and any of the changes observed in the translation rates largely corresponded to changes at the transcriptional level.

Given the lack of large-scale changes at the transcriptional and translational levels, but robust changes in cellular metabolism in the thiolation mutants, we focused our analysis on changes in expression levels of genes involved in metabolic pathways. We first examined several general amino

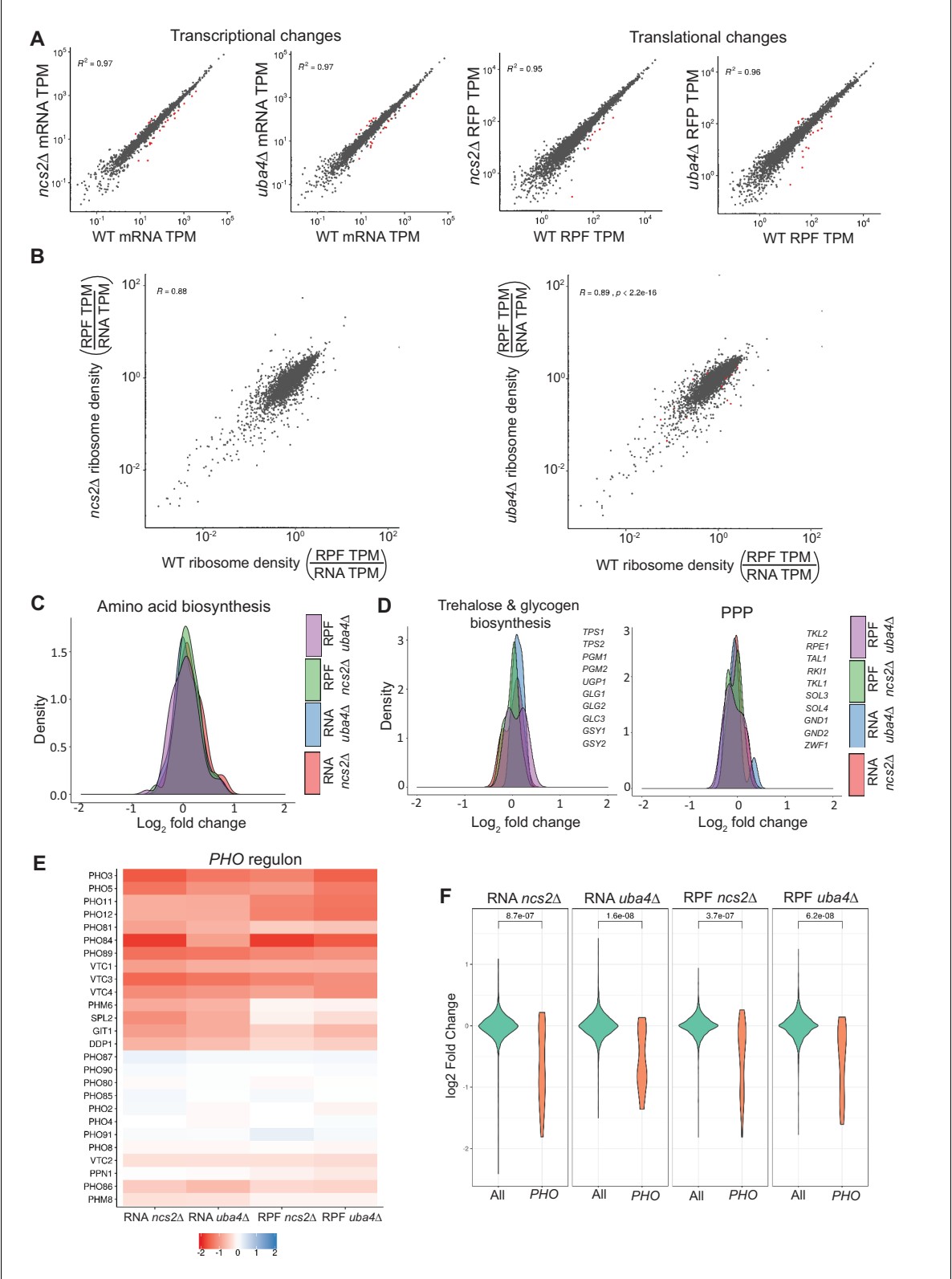

**Figure 4.** Loss of tRNA thiolation results in reduced phosphate homeostasis (*PHO*)-related transcripts and ribosome-footprints. (**A**) Correlation plots are shown, for WT and tRNA thiolation mutant cells (*ncs2Δ* and *uba4Δ*), for both gene expression (transcript) and ribosome footprint (translation) changes. The coefficients of determination ($R^2$) values are indicated, along with their significance (p values). Note: Very few differentially regulated genes were observed, and these are indicated as red points. Also see *Figure 4—figure supplement 1*. (**B**) Correlation plots comparing ribosome

*Figure 4 continued on next page*

*Figure 4 continued*

densities (RPF TPM/RNA TPM) for the thiolation mutants (*ncs2Δ* or *uba4Δ* respectively) with WT cells. Note: overall ribosome densities for thiolation mutants correlate exceptionally well with ribosome densities in WT cells, with the correlation coefficients (R) > 0.88 in both comparisons. Also see *Figure 4—figure supplement 1* for correlations between biological replicates of the same genetic background. (C) A density plot, representing changes in expression (transcript and ribosome footprints) of genes associated with amino acid biosynthetic pathways. In general, amino acid biosynthesis genes were upregulated in tRNA thiolation mutant cells (*uba4Δ* and *ncs2Δ*). Also see *Figure 4—figure supplement 3C*. (D) A density plot, representing changes in expression (transcript and ribosome footprints) of genes associated with trehalose and glycogen biosynthetic pathways, and the pentose phosphate pathway (PPP). In general, trehalose, glycogen biosynthesis and PPP genes were unchanged in tRNA thiolation mutant cells (*uba4Δ* and *ncs2Δ*), compared to WT cells. Also see *Figure 4—figure supplement 3C* (E) Heat map depicting changes in expression (transcript and ribosome footprints) of genes associated with the phosphate (*PHO*) regulon, in the tRNA thiolation mutant cells (*uba4Δ* and *ncs2Δ*), compared to WT cells, at the transcript and ribosome-footprint levels (log2 2-fold changes). (F) A density plot with a statistical comparison between WT and tRNA thiolation mutants, for changes in transcript or ribosome footprints of all genes in the genome, or only the *PHO* regulon. Note: $p < 10^{-7}$ for all the comparison datasets. Also see *Figure 4—figure supplement 3C*.

DOI: https://doi.org/10.7554/eLife.44795.011

The following figure supplements are available for figure 4:

**Figure supplement 1.** Transcript and ribosome footprint profiles in wild-type and tRNA thiolation mutant cells.

DOI: https://doi.org/10.7554/eLife.44795.012

**Figure supplement 2.** Gene expression (transcript) and ribosome footprint (translation) changes for wild-type (WT) and tRNA thiolation mutant cells.

DOI: https://doi.org/10.7554/eLife.44795.013

**Figure supplement 3.** Gene expression and ribosome footprint changes for wild-type (WT) and tRNA thiolation mutant cells in PHO regulon, ribosomal subunits, PPP, trehalose/glycogen, nucleotide and amino acid synthesis genes.

DOI: https://doi.org/10.7554/eLife.44795.014

acid control (GAAC) response genes, including the Gcn4 targets in amino acid biosynthetic pathways. Expectedly, we found these to be transcriptionally upregulated in the thiolation deficient cells (*Figure 4C*, and *Figure 4—figure supplement 3C*). Additionally, as expected, we observed increase in *GCN4* translation itself in the thiolation mutants, with no change in *GCN4* transcripts (*Figure 4—figure supplement 2B and C*). These data collectively corroborate our earlier data from *Figure 1*, and agrees with previous reports (*Zinshteyn and Gilbert, 2013*; *Nedialkova and Leidel, 2015*). Also consistent with our earlier data, most nucleotide biosynthesis genes showed an increase in mRNA and ribosome-footprint abundances in the thiolation deficient cells (*Figure 4—figure supplement 3A*; *Figure 4—figure supplement 3C*). In these datasets, there were also no obvious changes in central carbon metabolism genes in thiolation mutants. Notably, genes related to either trehalose/glycogen biosynthesis, or the PPP, showed negligible changes in either mRNA or ribosome-footprint abundances in the thiolation mutants (*Figure 4D*, and *Figure 4—figure supplement 3C*). This further reiterates that the rewiring observed in thiolation deficient cells was largely driven by metabolic flux. Finally, we also found small decreases in the transcription and translation rates of all large and small subunit genes of ribosomes in *uba4Δ* and *ncs2Δ* cells at the translational level (*Figure 4—figure supplement 3B and C*), consistent with earlier observations (*Laxman et al., 2013*; *Nedialkova and Leidel, 2015*).

However, in the course of this extensive functional analysis, we observed that an unusual group of ~20 genes were strongly downregulated in the thiolation mutants, both at the transcript and ribosome-footprint levels (*Figure 4E*). Although these do not obviously group into a single category based on gene-ontology (GO), we noted that these were functionally related to an important metabolic node. These genes are all part of the *PHO* regulon, which regulates phosphate homeostasis in cells (*Ljungdahl and Daignan-Fornier, 2012*; *Secco et al., 2012*). This downregulation of these *PHO*-related genes was exceptionally significant ($p < 10^{-7}$), compared to other genes across the genome, both at the level of transcript abundances and ribosome-footprints (*Figure 4E*, *Figure 4F*, and *Figure 4—figure supplement 3C*). Also notably, the only unaltered gene transcripts/ribosome footprints in the *PHO* regulon in the thiolation mutants, viz. *PHO2*, *PHO4*, *PHO80*, *PHO85*, *PHO87*, *PHO90* and *PHO91*, are transcription factors, cyclins/cyclin-dependent kinases or low affinity phosphate transporters that are not transcriptionally/translationally regulated, but are regulated at the level of their activity (*Lemire et al., 1985*; *Toh-e and Shimauchi, 1986*; *Madden et al., 1988*; *Yoshida et al., 1989*; *Madden et al., 1990*; *Schneider et al., 1994*; *Ogawa et al., 1995*; *Lenburg and O'Shea, 1996*; *Auesukaree et al., 2003*). Finally, we also observed that some genes related to phospholipid metabolism were downregulated in the thiolation mutants (*Figure 4—figure*

*supplement 3C*). Collectively, these data unexpectedly revealed a strong downregulation of genes related to phosphate homeostasis in the tRNA thiolation mutants.

## tRNA thiolation mutants exhibit a dampened *PHO* response

Inorganic phosphate (Pi) homeostasis is complex, but critical for overall nutrient homeostasis (*Ljungdahl and Daignan-Fornier, 2012*; *Secco et al., 2012*). The *PHO* regulon comprises of several genes that respond to phosphate starvation, and maintains internal phosphate levels by balancing transport of Pi from the external environment, from within vacuolar stores, and the nucleus (*Figure 5A*) (*Ljungdahl and Daignan-Fornier, 2012*; *Secco et al., 2012*). Extensive studies have defined global cellular responses to phosphate limitation (*Ogawa et al., 2000*; *Wykoff and O'Shea, 2001*; *Boer et al., 2003*; *Boer et al., 2010*; *Saldanha et al., 2004*; *Gresham et al., 2011*; *Levy et al., 2011*; *Choi et al., 2017*; *Gurvich et al., 2017*). In general, the *PHO* response is very sensitive to phosphate limitation, and is induced rapidly to restore internal phosphate levels. Extended phosphate starvation switches cells to an overall metabolically starved state. Our observed reduction in *PHO*-related transcripts and ribosome-footprints in the tRNA thiolation mutants was striking. We therefore first biochemically validated our results from the ribosome-profiling data, to more systematically investigate the extent of *PHO* downregulation. For this, we first measured protein amounts of Pho12 and Pho84 (two arbitrarily selected *PHO* genes which are downregulated in the thiolation mutants, as shown in Figure 4E), in WT cells and thiolation mutants grown in the same conditions used for ribosome profiling analysis. Amounts of these proteins were substantially reduced in *uba4Δ* and *ncs2Δ* cells (*Figure 5B*). We further compared Pho12 and Pho84 protein levels in WT cells and thiolation mutants under conditions of phosphate-limitation. Interestingly, we observed that even under these *PHO* inducible conditions, thiolation mutants exhibited substantially reduced levels of these *PHO* proteins compared to WT cells (*Figure 5C* and *Figure 5—figure supplement 1*), although the *PHO* response was induced. This suggested a constitutive dampening (but not shutdown and absence) of the *PHO* response in the thiolation mutants.

In order to more quantitatively estimate this dampening of the *PHO* response in the thiolation mutants, we utilized a robust assay to measure acid phosphatase activity. This represents the enzymatic activity of the *PHO* acid phosphatases, including Pho5, Pho11, Pho12 and the more constitutively expressed Pho3. Using this assay, we observed significantly reduced *PHO* activity in thiolation mutants (*Figure 5D*). Notably, this also quantitatively revealed that the thiolation mutants are unlike cells lacking Pho4 (*PHO* induction absent), or lacking Pho80 (constitutively extremely high *PHO* induction). These data reveal that the thiolation mutants are effectively in a constitutively phosphate-limited state, due to a dampened *PHO* response. These cells will therefore have a constitutively altered phosphate homeostasis, with reduced phosphate availability.

## Phosphate depletion in wild-type cells phenocopies tRNA thiolation mutants

We therefore asked if phosphate starvation in WT cells itself resembles the metabolic hallmarks of the thiolation mutants. We first biochemically estimated amounts of trehalose in WT cells with or without phosphate starvation, and found a robust increase in trehalose upon phosphate starvation (*Figure 6A*), much like the thiolation mutants. We next measured Gcn4 (protein) in WT cells, with or without phosphate starvation. Here, we observed a strong induction in Gcn4 protein upon phosphate starvation (*Figure 6B*). Further, like the Gcn4 response in the thiolation mutants, the Gcn4 induction in phosphate-starved WT cells was Gcn2-dependent (*Figure 6—figure supplement 1*). In addition to these data shown here in WT cells with phosphate starvation, earlier studies in response to phosphate limitation, have also observed high amino acid and low nucleotide levels under these conditions (*Boer et al., 2010*; *Klosinska et al., 2011*). These data are strikingly consistent with the metabolic profile observed in thiolation mutants, and suggests that the induction of Gcn4 observed in response to phosphate limitation might be due to reduced nucleotide levels. Notably, a metabolic signature of phosphate starvation is a depletion in ATP levels (*Boer et al., 2010*). We have already demonstrated not just reduced ATP levels in the thiolation mutants, but carbon flux through nucleotide synthesis, including ATP synthesis was also reduced in thiolation mutants (shown earlier in *Figure 2*). Finally, prior studies with phosphate limitation have shown that ribosomal genes are slightly repressed with phosphate limitation (*Saldanha et al., 2004*). In the thiolation mutants, where

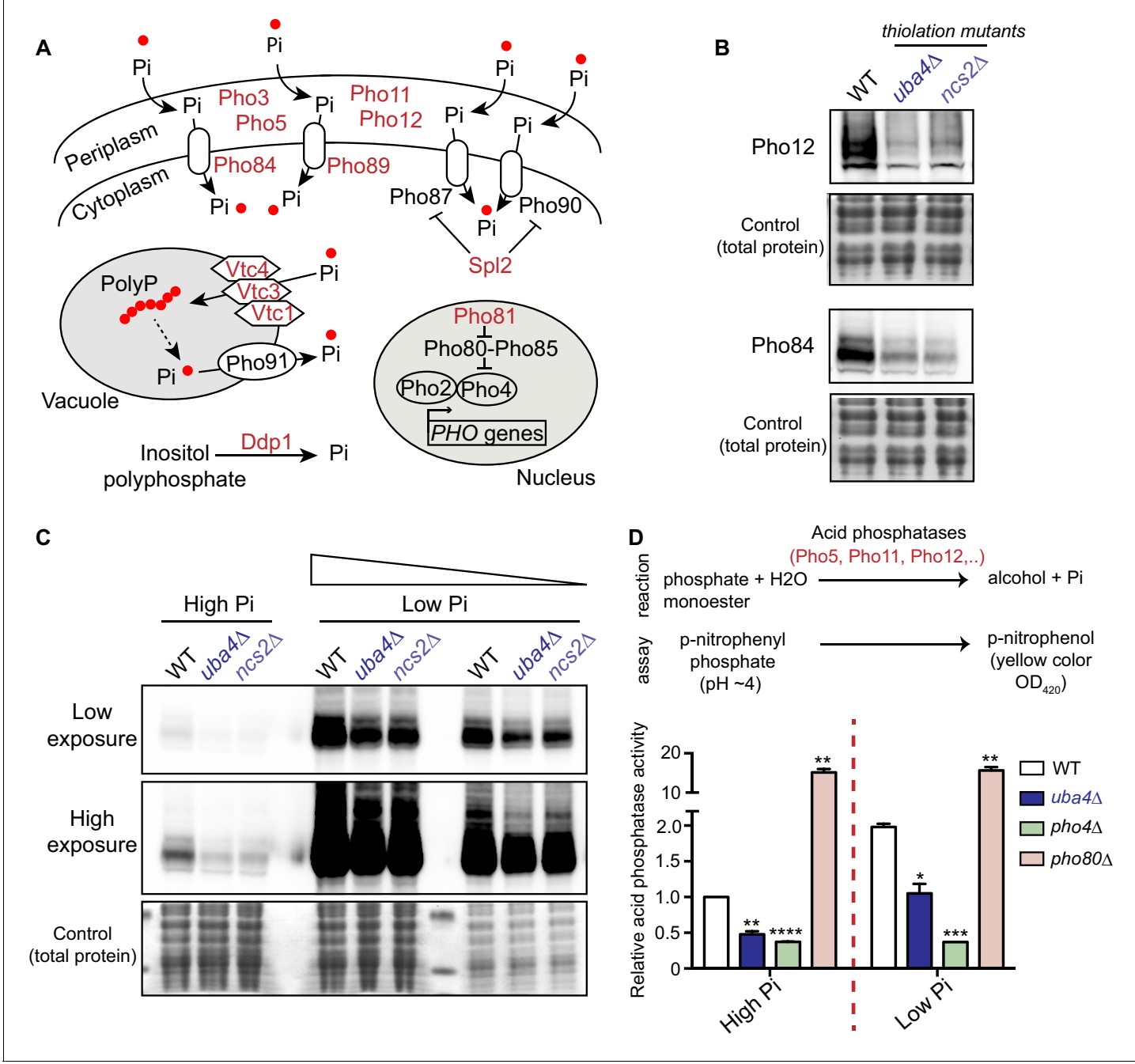

**Figure 5.** tRNA thiolation mutants exhibit a dampened *PHO* response. (**A**) A schematic representation of *PHO* regulon-related genes, and their roles. Pho84 and Pho89 are high affinity phosphate transporters, Pho87 and Pho90 are low affinity phosphate transporters, Spl2 is a negative regulator of low affinity phosphate transporters, Pho3, Pho5, Pho11 and Pho12 are secreted acid phosphatases, Pho80-Pho85 is a cyclin-dependent kinase (CDK) complex, Pho81 is a CDK inhibitor, Pho2 and Pho4 are transcription factors, Vtc1, Vtc3 and Vtc4 are involved in vacuolar polyphosphate accumulation, Pho91 is a vacuolar phosphate transporter, Ddp1 and Ppn1 are polyphosphatases. Proteins highlighted in red are down-regulated in tRNA thiolation mutant cells (*uba4Δ* and *ncs2Δ*). (**B**) Pho12 and Pho84 proteins are decreased in tRNA thiolation mutants. Pho12 and Pho84 protein levels (Pho12 and Pho84 tagged with FLAG epitope at their endogenous loci) in WT and tRNA thiolation mutant cells (*uba4Δ* and *ncs2Δ*) grown in minimal media were detected by Western blot analysis using an anti- FLAG antibody. A representative blot is shown (n = 3). (**C**) Pho84 protein is decreased in tRNA thiolation mutants in both high and low Pi conditions. Pho84 protein levels (the carboxy-terminus of Pho84 tagged with a FLAG epitope at its endogenous locus) in WT and tRNA thiolation mutant cells (*uba4Δ* and *ncs2Δ*) grown in high and low Pi media were detected and compared by Western blot analysis using an anti- FLAG antibody. For high and low Pi condition comparisons, the same amount of total protein was loaded in each lane of the SDS-PAGE gel. For low Pi, two different concentrations of total protein (undiluted and 1:3 diluted) were loaded, since the *PHO*-related proteins are strongly induced in this condition, and hence a dilution is required to visualize a non-saturated image. A representative blot is shown

*Figure 5 continued on next page*

*Figure 5 continued*

(n = 2). Also see *Figure 5—figure supplement 1A*. (**D**) Acid phosphatase activity is decreased in tRNA thiolation mutants. A schematic representation of the acid phosphatase reaction, and the colorimetric assay used to study this reaction, is shown. This quantitatively measures the collective intracellular activity of the *PHO* acid phosphatases Pho5, Pho11, Pho12 and Pho3. Acid phosphatase activity was determined in WT, tRNA thiolation mutant (*uba4Δ*), *pho4Δ* and *pho80Δ* in both high and low Pi media conditions, using this assay. Acid phosphatase activity was plotted relative to WT grown in high Pi, which was set to 1. * denotes statistical significance (Student's t-test), comparing all samples to WT in both high and low Pi condition. Data are displayed as means ± SD, n = 2 biological replicates with three technical replicates each. *p<0.05, **p<0.01, ***p<0.001, ****p<0.0001.
DOI: https://doi.org/10.7554/eLife.44795.015
The following figure supplement is available for figure 5:

**Figure supplement 1.** tRNA thiolation mutants exhibit a dampened *PHO* response.
DOI: https://doi.org/10.7554/eLife.44795.016

phosphate homeostasis is affected due to the dampened *PHO* response, we also observed a small decrease in ribosomal genes (as shown earlier in *Figure 4—figure supplement 3B*; *Figure 4—figure supplement 3C*). Thus, these data from WT cells starved of phosphate strikingly phenocopied the tRNA thiolation mutants.

Finally, we used the acid phosphatase activity assay (described earlier) as a robust read-out for the extent of the *PHO* response, to compare activities in WT cells, cells lacking tRNA thiolation, and cells lacking Gcn4 (individually or in combination). This was done to test if the Gcn4 induction was upstream or downstream of the *PHO* response. Notably, the loss of *GCN4* in cells lacking thiolation (*uba4Δ gcn4Δ*) also resulted in significantly decreased acid phosphatase activity (*Figure 6C*), similar to the thiolation mutants. However, the loss of *GCN4* alone had no effect on acid phosphatase

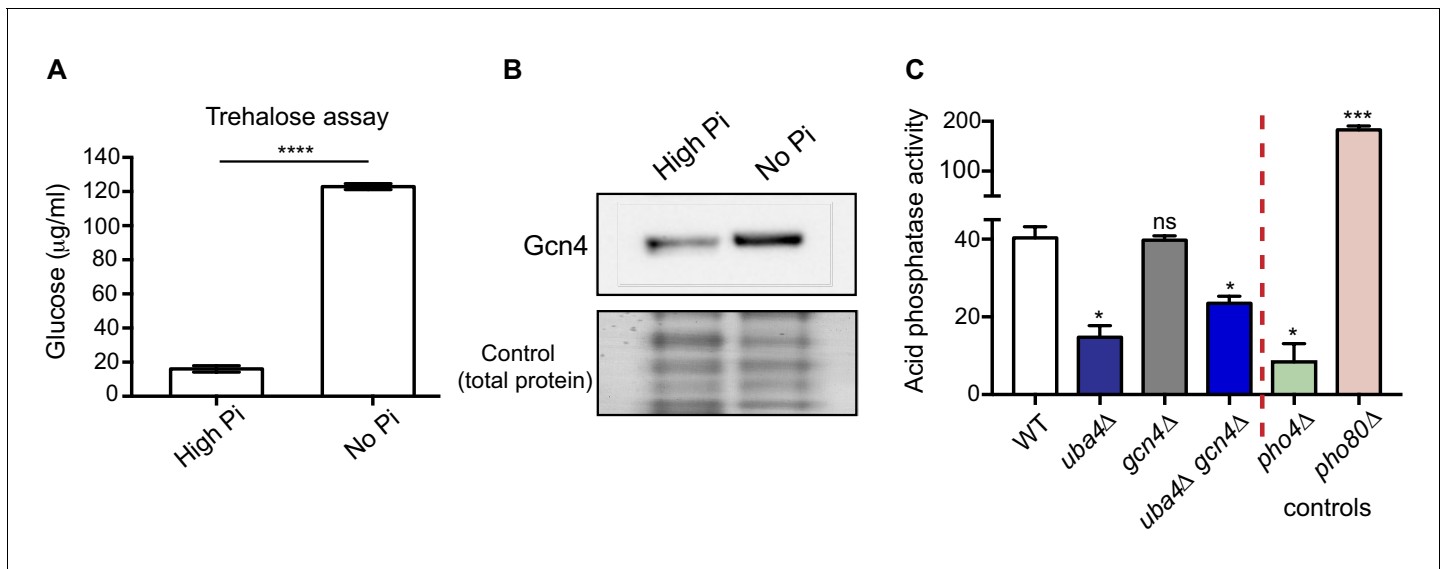

**Figure 6.** Phosphate depletion in wild-type cells phenocopies tRNA thiolation mutants. (**A**) Trehalose amounts are increased upon phosphate starvation. Trehalose content of WT cells grown in high and no Pi media was plotted. Data are displayed as means ± SD, n = 3. ****p<0.0001. (**B**) Gcn4 protein is increased upon phosphate starvation. Gcn4 protein levels (Gcn4 tagged with HA epitope at the endogenous locus) in WT grown in high and no Pi media were detected by Western blot analysis using an anti-HA antibody. A representative blot is shown (n = 3). (**C**) Cells lacking Uba4 and Gcn4 (*uba4Δ gcn4Δ*) also show decreased acid phosphatase activity. Acid phosphatase activity was determined in WT, tRNA thiolation mutant (*uba4Δ*), *gcn4Δ*, *uba4Δ gcn4Δ*, *pho4Δ* and *pho80Δ* grown in high Pi media conditions supplemented with amino acids, by a colorimetric assay. Note that loss of Gcn4 alone has no effect on acid phosphatase activity. * denotes statistical significance (Student's t-test), comparing all samples to WT. Data are displayed as means ± SD, n = 3 biological replicates (with three technical replicates each). ns denotes non-significant difference. *p<0.05, ***p<0.001.
DOI: https://doi.org/10.7554/eLife.44795.017
The following figure supplement is available for figure 6:

**Figure supplement 1.** Gcn2-dependent translational induction of Gcn4 during phosphate starvation.
DOI: https://doi.org/10.7554/eLife.44795.018

activity. This suggests that the Gcn4 induction is downstream of the dampened *PHO* response in the thiolation mutants.

Collectively, these results strongly suggest that effective phosphate limitation is responsible for the metabolic state switch exhibited by the thiolation deficient cells.

## Trehalose synthesis associated phosphate release enables cells to maintain phosphate balance

This observed downregulation of phosphate metabolism in the thiolation deficient cells is striking. Nonetheless, it is not immediately obvious biochemically how this relates to re-routing carbon towards storage carbohydrates, and decoupling amino acid metabolism from nucleotide synthesis. Perplexingly, in our transcript and translation analysis, no other metabolic arms were similarly decreased in the thiolation deficient cells, and only the amino acid biosynthesis arm (dependent on Gcn4) increases, which we have addressed earlier. Notably, while earlier studies have hinted that phosphate limitation results in a shift towards storage carbohydrates (*Lillie and Pringle, 1980*; *Boer et al., 2003*; *Boer et al., 2010*), this more extensive metabolic rewiring has not been carefully analyzed, and a biochemical explanation for this is missing. We wondered if some overlooked aspect within this biochemical process could explain why a perturbation in phosphate homeostasis connects to the synthesis of storage carbohydrates trehalose and glycogen, as is also seen in tRNA thiolation mutants. To address this, we carefully examined all the metabolic nodes altered in the tRNA thiolation mutants, evaluating necessary co-factors and products of each pathway, and looking for possible connections to phosphate. Here, we noted an apparently minor, largely ignored output in the arm of carbon metabolism, where glucose-6-phosphate is routed towards trehalose synthesis. The first step of trehalose synthesis is the formation trehalose-6-phosphate (T-6-P), carried out by trehalose-6-phosphate synthase (Tps1). This is followed by the dephosphorylation of T-6-P by Tps2 (*De Virgilio et al., 1993*), forming trehalose (*Figure 7A*). We noted that this Tps2-dependent second step is accompanied by the release of free, inorganic phosphate (Pi) (*Figure 7A*). Canonically, these two steps are viewed as an apparently futile trehalose cycle during glycolysis, regenerating glucose, in order to maintain balanced glycolytic flux (*Heerden et al., 2014*; *van Heerden et al., 2014*). However, we reasoned that if the availability of inorganic phosphate is limiting, a shift to trehalose synthesis can be a way by which cells can liberate Pi, and restore phosphate levels. For this to be generally true, the prediction is that during phosphate starvation, WT cells must accumulate trehalose in order to recover phosphate. As shown earlier, this is exactly what is observed in WT cells limited for phosphate (*Figure 6A*), and in the tRNA thiolation mutants (*Figure 2E*; *Figure 2F*) which are effectively phosphate limited due to a reduction in the *PHO* genes.

Given the central role of phosphate, cells utilize all means possible to restore internal phosphate (*Ljungdahl and Daignan-Fornier, 2012*). Therefore it is experimentally challenging to study changes in phosphate homeostasis in cells. However, we directly tested the hypothesis that trehalose synthesis is a direct way for cells to restore internal phosphate in tRNA thiolation mutants, by utilizing cells lacking *TPS2*. These cells cannot complete trehalose synthesis, and importantly cannot release phosphate (*Figure 7A*). We first measured the intracellular Pi levels in WT cells, thiolation mutants (*uba4Δ*), cells lacking Tps2p (*tps2Δ*), and cells lacking tRNA thiolation as well as Tps2p (*uba4Δ tps2Δ*). *pho85Δ* cells were used as a control, since they exhibit intrinsically higher intracellular Pi levels (*Liu et al., 2017*). We first observed that in cells lacking *TPS2* (*tps2Δ*) intracellular Pi levels were lower (~75–80%) relative to WT cells (*Figure 7B* and *Figure 7—figure supplement 1A*). This suggests that while other pathways (phosphate uptake, glycerol production and vacuolar phosphate export) remain relevant, Tps2p-mediated Pi release by dephosphorylation of trehalose-6-P is itself important for maintaining internal phosphate levels. Importantly, *uba4Δ* cells had only slightly reduced intracellular Pi levels (~90%) relative to WT cells (*Figure 7B* and *Figure 7—figure supplement 1A*). This is consistent with the prediction that due to reduced *PHO* expression in these cells, phosphate homeostasis is altered, but they can compensate for phosphate availability through increased trehalose synthesis. Contrastingly, the cells lacking both thiolation and Tps2 (*uba4Δ tps2Δ*) showed a dramatic reduction in Pi levels (~65%), compared to either of their single mutants. This striking reduction in Pi levels in these cells is consistent with the predicted outcome, where an inability to release phosphate from trehalose (*tps2Δ*) is also coupled with reduced expression of phosphate assimilation genes (*uba4Δ*). Next, we tested possible genetic interactions between *tps2Δ* and thiolation mutants (*uba4Δ* and *ncs2Δ*) by assessing relative growth. In our genetic background, *tps2Δ*

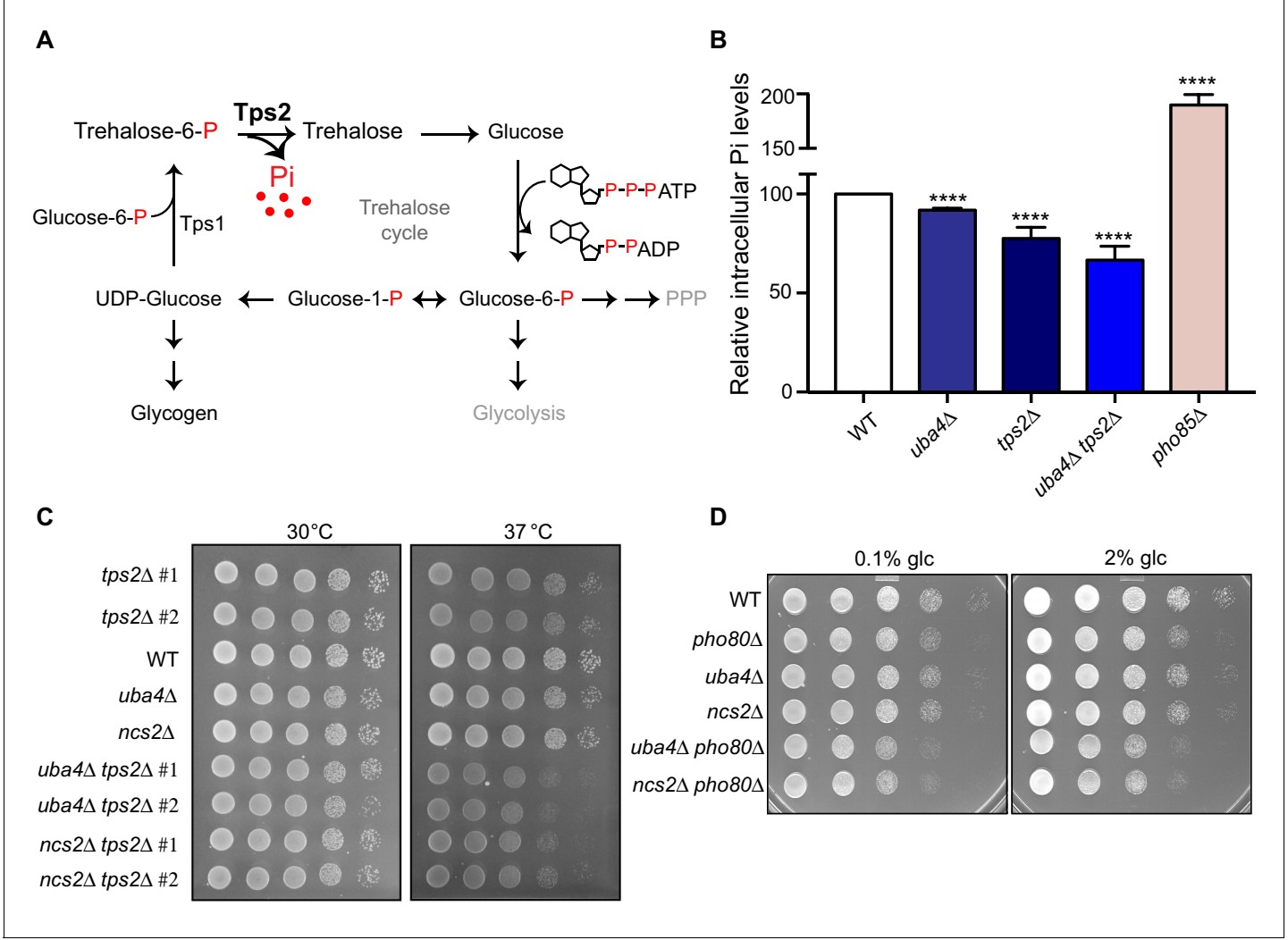

**Figure 7.** Trehalose synthesis associated phosphate release enables cells to maintain phosphate balance. (**A**) A schematic representation of the trehalose cycle, showing the routing of glucose-6-phospate towards trehalose and glycogen biosynthesis. Trehalose synthesis requires two glucose molecules, catalysed by the Tps1 and Tps2 enzymes. The Tps2-mediated reaction synthesizes trehalose, and notably releases free Pi. (**B**) Intracellular Pi levels are maintained in tRNA thiolation mutant cells (*uba4Δ*) by Tps2 activity. Free intracellular Pi levels were determined in WT, tRNA thiolation mutant (*uba4Δ*), *tps2Δ*, *uba4Δ tps2Δ* and *pho85Δ* cells grown in minimal media, by a colorimetric assay. Intracellular Pi levels in mutant cells relative to WT are plotted, where WT was set to 100. Data are displayed as means ± SD, at least n = 2 biological replicates with three technical replicates each. ****p<0.0001, Student's t-test, comparing all samples to WT. Also see *Figure 7—figure supplement 1A*. (**C**) Synthetic genetic interaction between *TPS2* and tRNA thiolation genes (*UBA4* and *NCS2*) at 37° C. WT, tRNA thiolation mutants (*uba4Δ* and *ncs2Δ*), *tps2Δ*, *uba4Δ tps2Δ* and *ncs2Δ tps2Δ* double mutant cells grown in minimal media were spotted on low Pi media agar plates with 2% glucose and incubated at 30° C and 37° C. Also see *Figure 7—figure supplement 1B*. (**D**) Synthetic genetic interaction between *PHO80* and tRNA thiolation genes (*UBA4* and *NCS2*). WT, tRNA thiolation mutants (*uba4Δ* and *ncs2Δ*), *pho80Δ*, *uba4Δ pho80Δ* and *ncs2Δ pho80Δ* double mutant cells grown in minimal media (0.1% glucose) were spotted on minimal media agar plates (0.1% and 2% glucose) and incubated at 30° C.

DOI: https://doi.org/10.7554/eLife.44795.019

The following figure supplement is available for figure 7:

**Figure supplement 1.** Tps2 maintains phosphate balance in tRNA thiolation deficient cells.
DOI: https://doi.org/10.7554/eLife.44795.020

cells exhibit slightly slower growth at 37°C. Notably, in cells lacking both *TPS2* and tRNA thiolation (*tps2Δ uba4Δ* or *tps2Δ ncs2Δ*), we observed a strong synthetic growth defect, in conditions of low phosphate as well as normal phosphate (*Figure 7C* and *Figure 7—figure supplement 1B*). This is entirely consistent with the proposed role of Tps2p in maintaining phosphate balance in thiolation mutants. Finally, if we completely imbalance phosphate homeostasis in cells, using cells lacking

*PHO80*, individual mutants of either *pho80Δ* or thiolation deficient cells show minimal growth defects, but double mutants (*pho80Δ uba4Δ* or *pho80Δ ncs2Δ*) show a severe synthetic growth defect (*Figure 7D*).

Collectively, our results suggest that altering phosphate homeostasis by decreasing *PHO* activity regulates overall carbon and nitrogen flow. Cells can therefore deal with decreased phosphate availability by diverting carbon flux away from nucleotide biosynthesis, and towards Tps2-dependent trehalose synthesis and Pi release. This restores phosphate, while concurrently resulting in an accumulation of amino acids, and a reduction in nucleotide synthesis.

## Discussion

In this study, we highlight two related findings- a direct role for a component of translational machinery, $U_{34}$ thiolated tRNAs, in regulating cellular metabolism by controlling phosphate homeostasis; and a biochemical rationale for how phosphate availability regulates flux through carbon and nitrogen metabolism.

An integrative model emerges from our studies, explaining how high amounts of thiolated tRNAs reflect a 'growth state', while reduced tRNA thiolation reflect a 'starvation state' (*Figure 8*). Cells can use tRNA thiolation to sense overall nutrient sufficiency and appropriately modulate metabolic outputs, using phosphate homeostasis as the metabolic control point (*Figure 8*). In this model, tRNAs are thiolated in tune with methionine and cysteine availability, as has been demonstrated earlier (*Laxman et al., 2013*). Separately, the presence of these sulfur amino acids themselves reflect an overall amino acid sufficiency state, leading to an anabolic program capable of sustaining cell growth and proliferation (*Walvekar et al., 2018b*). Methionine up-regulates both amino acid synthesis and carbon flux leading towards nucleotide synthesis, and therefore growth (*Walvekar et al., 2018b*). Collectively therefore, in conditions of methionine sufficiency (and therefore amino acid sufficiency), where tRNAs are maximally thiolated, cells direct carbon flux towards nucleotide biosynthesis, coupled with amino acid utilization (as shown in *Figures 1* and *2*). Accordingly, at this level of metabolic coupling, thiolated tRNAs sense amino acids, and ensure appropriate nucleotide levels for growth and cell cycle progression (as shown in *Figures 2* and *3*). On the other hand, the loss of thiolated tRNAs results in an inability of cells to fully sense and integrate these nutrient cues, rewiring carbon and nitrogen flux away from nucleotide synthesis and instead towards storage carbohydrates. This switches cells to a 'starvation-like state'. This metabolic rewiring mediated by tRNA thiolation is achieved not by directly regulating enzymes in these arms of carbon metabolism, but instead by down-regulating the *PHO* regulon. This constricts intracellular phosphate availability (as shown in *Figures 4* and *5*). A result of this dampened *PHO* response, and constriction in available free phosphate, is that in order to restore phosphate, cells divert glucose flux towards Tps2-mediated trehalose synthesis, which concurrently releases Pi (as shown in *Figures 6* and *7*). Thus, while the trehalose shunt and phosphate recycling restores phosphate levels, this is at the cost of decreased nucleotide biosynthesis, and delayed cell cycle progression. Effectively, the loss of tRNA thiolation rewires cells to a starved metabolic state. Collectively, tRNA thiolation appropriately regulates metabolic outputs by controlling phosphate homeostasis, thereby enabling cells to commit to growth (*Figure 8*). Intriguingly, this correlation of tRNA thiolation with growth and rewired metabolism is emerging in cancer development (*McMahon and Ruggero, 2018*; *Rapino et al., 2018*), suggesting possibly conserved metabolic roles for these modified tRNAs.

Insight into a deeper coupling of discrete metabolic arms emerges from this study, suggesting how cells can fully integrate carbon, nitrogen, sulfur and phosphate inputs for optimal growth. Earlier studies have noted a strong correlation of phosphate starvation with decreased nucleotide synthesis, and ATP availability (*Boer et al., 2010*; *Klosinska et al., 2011*). Reduced ATP synthesis is a hallmark of phosphate starvation (*Boer et al., 2010*), along with increased trehalose synthesis (discussed in a subsequent paragraph). Through this metabolic rewiring, reduced flux through the PPP and one-carbon/folate cycle can be inferred. A reduction in these metabolic pathways will reduce not just nucleotide synthesis, but also the production of NADPH, and cellular reductive biosynthetic capacity (*Fan et al., 2014*; *Hosios and Vander Heiden, 2018*). Notably, the reductive costs in terms of NADPH utilized to assimilate sulfates into sulfur amino acids (methionine and cysteine), and their subsequent metabolites, are themselves extremely high (*Thomas and Surdin-Kerjan, 1997*; *Kaleta et al., 2013*; *Walvekar et al., 2018b*). Indeed, this coupling of NADPH production and

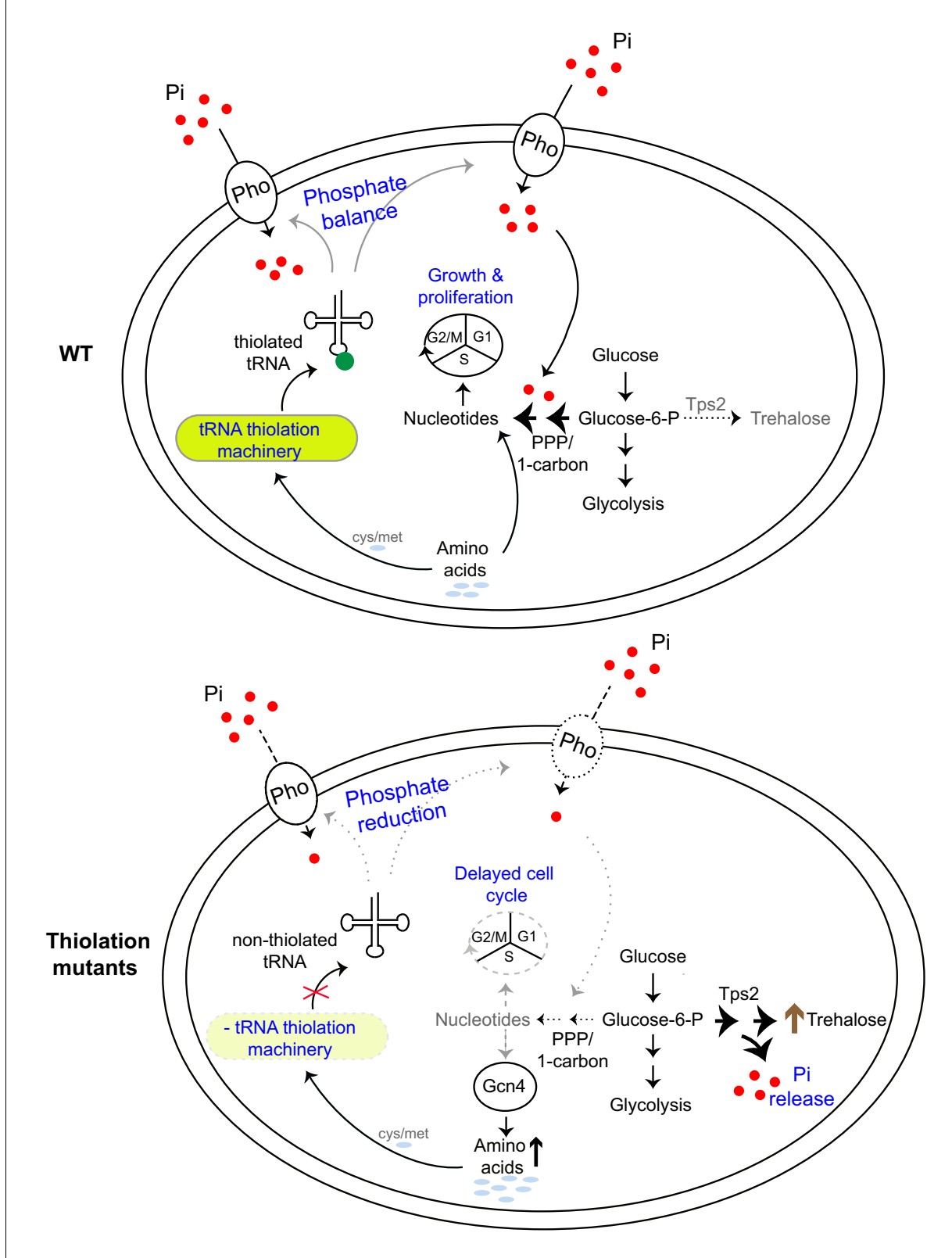

**Figure 8.** A model illustrating how tRNA thiolation regulates the metabolic state of the cell. WT cells (which have a functional tRNA thiolation machinery) in amino acid (methionine and cysteine) replete conditions have high amounts of thiolated tRNAs. In these cells, carbon flux coupled with amino acid utilization is towards nucleotide biosynthesis. Sufficient nucleotide levels support growth, and proper cell cycle progression, and reflect an overall 'growth' metabolic state. The absence of the tRNA thiolation machinery results in non-thiolated tRNAs, and here carbon flux is driven away from

*Figure 8 continued on next page*

*Figure 8 continued*

nucleotide synthesis and towards storage carbohydrates, with a concurrent accumulation of amino acids. This metabolic rewiring in these mutants is due to reduced expression of genes involved in phosphate-responsive signalling pathway (a dampened *PHO* response), which results in a phosphate-limited state. Due to this restricted phosphate availability in the thiolation mutants, these cells attempt to restore intracellular phosphate levels through a metabolic rewiring, via Tps2-dependent trehalose synthesis, accompanied by phosphate release. Thus, despite the absence of carbon or nitrogen (amino acid) starvation, the tRNA thiolation mutants exhibit an overall metabolic signature of a 'starved state', with decreased nucleotide synthesis and delayed cell cycle progression. Key: Pho: phosphate transporters, Pi: inorganic phosphate, PPP: Pentose Phosphate Pathway, cys: cysteine, met: methionine.

DOI: https://doi.org/10.7554/eLife.44795.021

methionine availability is also observed in the converse direction (as noted earlier in the text), where the presence of methionine increases PPP flux (*Walvekar et al., 2018b*), and mutants in the PPP pathway are methionine auxotrophs (*Thomas et al., 1991*). Therefore, an amplified regulatory response in relation to tRNA thiolation can be imagined. In such a response, in the presence of methionine, there is an increase in PPP and one-carbon flux and nucleotide synthesis, which is amplified by the maximally thiolated tRNAs by ensuring sufficient phosphate availability and an intact *PHO* response. Conversely, reduced sulfate assimilation in turn could limit thiolation, and thereby amplify the regulatory response, again using phosphate homeostasis as a control point. Indeed, the metabolic state of the tRNA thiolation mutants (and the phenotypes associated) are consistent with what is expected in a cell with reduced NADPH availability and reduced reductive biosynthetic capacity. Our data suggest a subtler, more integrative role for tRNA thiolation in optimally sensing overall nutrient sufficiency, and modulating overall metabolic responses leading to a growth-sustaining state.

More generally, our findings identify a biochemical reaction, the trehalose shunt, that explains how phosphate homeostasis determines the extent of carbon and nitrogen flux towards nucleotide synthesis. While it is textbook knowledge that inorganic phosphate is important for glucose homeostasis (*Mason et al., 1981*; *Boyle, 2005*; *Heerden et al., 2014*; *van Heerden et al., 2014*), the biochemical connection of phosphate balance to carbon and nitrogen flux remains poorly explained. Studies have observed that trehalose increases upon phosphate starvation (*Lillie and Pringle, 1980*; *Klosinska et al., 2011*), and *TPS2* is upregulated (*Ogawa et al., 2000*). In these conditions, central carbon metabolism is down, and phosphate limitation is a 'general starvation' cue (*Brauer et al., 2008*; *Boer et al., 2010*; *Gurvich et al., 2017*). Why this occurs has not been immediately apparent. Here, identifying the trehalose shunt as a way to restore phosphate balance, explains these observations. These data also explain earlier observations from pathogenic fungi, showing that that trehalose synthesis determines flux through the pentose phosphate pathway, and nitrogen metabolism (*Wilson et al., 2007*). Additionally, phosphate starvation results in better cell survival in limited nutrient conditions, and also promotes efficient recovery when nutrients become available (*Gurvich et al., 2017*). Since trehalose accumulation and utilization are respectively tightly coupled with exit from and re-entry into the cell division cycle (*Shi et al., 2010*; *Shi and Tu, 2013*), we propose a dual role for trehalose synthesis during phosphate-limitation. During phosphate limitation, trehalose synthesis concurrently releases inorganic phosphate, which restores phosphate balance in the cell and diverts flux away from nucleotide synthesis and growth. When phosphate is no longer limiting, cells can liquidate trehalose to re-enter the cell division cycle, enabling rapid recovery. Thus, this study adds an important biochemical function to the many roles played by this versatile metabolite. These include cell survival during desiccation and freezing (*Calahan et al., 2011*; *Erkut et al., 2011*; *Erkut et al., 2016*; *Tapia and Koshland, 2014*; *Tapia et al., 2015*), the ability to act as a protein chaperone (*Tapia and Koshland, 2014*), and as a membrane protectant (*Abusharkh et al., 2014*).

A modified tRNA is an unusual but effective mechanism to coordinately integrate sensing of overall nutrient sufficiency, and regulate metabolic homeostasis. While previous studies have observed decreased phosphate-related transcripts in tRNA thiolation deficient cells (*Leidel et al., 2009*; *Nedialkova and Leidel, 2015*; *Chou et al., 2017*), possible roles for phosphate in tRNA thiolation mediated function have been ignored. Furthermore, tRNAs undergo other conserved modifications in the $U_{34}$ position: 5-methoxycarbonylmethyluridine (mcm$^5$ $U_{34}$), 2-thiouridine (s$^2$ $U_{34}$), 5-methoxycarbonylmethyl-2-thiouridine (mcm$^5$s$^2$ $U_{34}$), 5-methylaminomethyluridine (mnm$^5$U$_{34}$) (*Phizicky and*

*Hopper, 2010*). In this study we focus only on how the $s^2 U_{34}$ modification (which is derived from sulfur amino acids) regulates cellular metabolic state. Interestingly, the other $U_{34}$ modifications all require s-adenosyl methionine (SAM), and SAM is itself directly derived from sulfur amino acid metabolism (*Thomas and Surdin-Kerjan, 1997*. Mutants of all these related $U_{34}$ tRNA modifications show similar metabolic phenotypes as the thiolation mutants (*Zinshteyn and Gilbert, 2013*; *Nedialkova and Leidel, 2015*; *Chou et al., 2017*; *Han et al., 2018*), and have a down-regulated *PHO* response (*Chou et al., 2017*). This raises the possibility that these $U_{34}$-tRNA and other tRNA modifications derived from amino acid metabolism use similar mechanisms, of controlling phosphate availability in order to regulate metabolic homeostasis. A primordial role of such tRNA modifications might therefore be to appropriately sense overall amino acid sufficiency (with methionine as a sentinel growth signal; *Walvekar et al., 2018b*), and modulate metabolic states towards growth, regulating phosphate availability as a means to achieve this. Co-opting tRNAs (which are the translation components most closely linked to amino acids) to control metabolic states can therefore be an efficient means to ensure appropriate commitments to growth and proliferation, and maximize cellular fitness.

Concluding, here we discover that a sulfur amino acid-dependent tRNA modification (thiolated $U_{34}$) enables cells to appropriately balance amino acid and nucleotide levels and regulate metabolic state, by controlling phosphate homeostasis. More generally, we suggest how phosphate homeostasis can impact flux through different arms of carbon and nitrogen metabolism.

## Materials and methods

### Key resources table

| Reagent type (species) or resource | Designation | Source or reference | Identifiers | Additional information |
|---|---|---|---|---|
| Antibody | anti-HA | Roche | Cat# 12CA5 | WB (1:2000) |
| Antibody | anti-phospho eIF2a (Ser51) | Cell Signalling Technology | Cat# 9721S | WB (1:1000) |
| Antibody | anti-FLAG | Sigma-Aldrich | Cat # F1804-5MG | WB (1:2000) |
| Antibody | anti-Rad53 yC-19 | Santa Cruz Biotechnology | Cat # sc-6749 | WB (>1:1000) |
| Antibody | HRP-conjugated secondary antibodies (anti-mouse, anti-rabbit) | Sigma-Aldrich | Cat # 7076S Cat # 7074S | WB (1:5000) |
| HPLC column | Synergi 4µ Fusion-RP 80A column (100 × 4.6 mm) | Phenomenex | Cat # 00D-4424-E0 | |
| Sequence-based reagent | Gcn4-luciferase translation reporters | this study | | |
| Peptide, recombinant protein | trehalase | Sigma-Aldrich | Cat # T8778 | |
| Peptide, recombinant protein | amyloglucosidase | Sigma-Aldrich | Cat # 10115 | |
| Peptide, recombinant protein | RNAse A | Sigma-Aldrich | Cat # R4875 | |
| Peptide, recombinant protein | protease solution | Sigma-Aldrich | Cat # P6887 | |
| Commercial assay or kit | Glucose estimation kit | Sigma-Aldrich | Cat # GAGO20 | |
| Commercial assay or kit | RiboZero | Epicenter | Cat # MRZH116 | |
| Commercial assay or kit | luciferase assay kit | Promega | Cat # E1500 | |

*Continued on next page*

*Continued*

| Reagent type (species) or resource | Designation | Source or reference | Identifiers | Additional information |
|---|---|---|---|---|
| Commercial assay or kit | Maxima SYBR Green/ROX qPCR Master Mix | Thermo Scientific | Cat # K0222 | |
| Commercial assay or kit | ATP estimation kit | Thermo Scientific | Cat # A22066 | |
| Chemical compound, drug | SYTOX green | Thermo Scientific (Invitrogen) | Cat # S7020 | |
| Chemical compound, drug | p-nitrophenyl phosphate | Sigma-Aldrich | Cat # N4645 | |
| Chemical compound, drug | bicinchoninic acid assay | Thermo Scientific | Cat # 23225 | |
| Software, algorithm | Prism 7 | Graphpad | | |

## Yeast strains, media and growth conditions

The prototrophic CEN.PK strain of *Saccharomyces cerevisiae* was used in all the experiments (*van Dijken JP et al., 2000*). All the strains used in this study are listed in *Supplementary file 1*. For all experiments, cells were grown overnight at 30°C in rich media (1% yeast extract, 2% peptone, 2% dextrose), washed once and subsequently sub-cultured in minimal media (0.67% yeast nitrogen base with ammonium sulfate, without amino acids, 0.1% glucose) unless specified. Phosphate-limited medium was prepared as described previously (*Klosinska et al., 2011*) except that 0.1%. glucose was used unless specified. The only source of phosphorus in phosphate-limited media (low Pi) was $KH_2PO_4$, which was present at a concentration of 0.15 mM. In high phosphate media (high Pi), $KH_2PO_4$ was present at a concentration of 7.5 mM with 0.1%. glucose unless specified. In no phosphate media (no Pi), $KH_2PO_4$ was completely absent and 0.1%. glucose was used. Complete medium was high Pi media supplemented with amino acids and 0.1% glucose. Amino acid concentrations were used as described previously (*Sherman, 2002*). Sulfur-rich medium was minimal media (0.67% yeast nitrogen base with ammonium sulfate, without amino acids) with 2% glucose. Sulfur-starved medium was prepared as described previously with 2% glucose (*Kankipati et al., 2015*) Sulfur amino acid limited medium was minimal media (0.67% yeast nitrogen base with ammonium sulfate, without amino acids) supplemented with all amino acids at a final concentration of 2 mM with methionine and cysteine being completely absent and 2% glucose.

## Western blot analysis

For Gcn4, total eIF2$\alpha$, P- eIF2$\alpha$, Pho12, Pho84 and Rad53 protein levels, cells were grown overnight in rich media, washed once and subsequently sub-cultured in minimal media at an initial $OD_{600}$ of 0.1 and grown till the $OD_{600}$ reached 0.8–1.0. For Pho84 and Pho12 protein levels in high and low Pi media, cells were grown overnight in rich media, washed once and subsequently sub-cultured in high and low Pi media at an initial $OD_{600}$ of 0.1 and incubated for 5 hr at 30°C. For Gcn4 protein levels in high and no Pi media, cells were grown overnight in rich media, washed once and subsequently sub-cultured in high and no Pi media at an initial $OD_{600}$ of 0.1 and incubated for 8 hr at 30°C. Cells were harvested by centrifugation and protein was isolated by trichloroacetic acid (TCA) precipitation method. Briefly, cells were resuspended in 400 µl of 10% trichloroacetic acid and lysed by bead-beating three times. The precipitates were collected by centrifugation, resuspended in 400 µl of SDS-glycerol buffer (7.3% SDS, 29.1% glycerol and 83.3 mM Tris base) and heated at 100°C for 10 min. The lysate was cleared by centrifugation and protein concentration was determined by using a bicinchoninic acid assay (23225, Thermo Fisher). Equal amounts of samples were electrophoretically resolved on 4–12% pre-cast Bis-tris polyacrylamide gels (NP0322BOX, Invitrogen). Anti-HA (12CA5, Roche) was used to detect Gcn4-HA, anti-phospho eIF2$\alpha$ (Ser51) (9721S, Cell Signalling Technology) was used to detect phospho-eIF2$\alpha$, anti-FLAG (F1804-5MG, Sigma-Aldrich) was used to detect total eIF2$\alpha$ (eIF2$\alpha$-FLAG), Pho12-FLAG and Pho84-FLAG, anti-Rad53 yC-19 (sc-6749, Santa Cruz Biotechnology) was used to detect phosphorylated Rad53 protein. Horseradish peroxidase-conjugated

secondary antibodies (mouse and rabbit) were obtained from Sigma-Aldrich. For Western blotting, standard enhanced chemiluminescence reagent (GE Healthcare) was used. Coomassie brilliant blue R-250 was used to stain gels for loading control.

## Metabolite extraction and LC-MS/MS analysis

For steady state amino acids and nucleotides levels, cells were grown overnight in rich media, washed once and subsequently sub-cultured in minimal media at an initial $OD_{600}$ of 0.1 and grown till the $OD_{600}$ reached ~0.8.~10 $OD_{600}$ cells were quenched with 60% methanol at −40°C, and metabolites were extracted, as explained in detail elsewhere (*Walvekar et al., 2018a*). For steady state amino acids, nucleotides and sulfur-containing metabolite levels in sulfur-rich and sulfurstarved conditions, wild-type cells were grown overnight in rich media, washed once and subsequently sub-cultured in sulfur-rich and sulfur-starved conditions at an initial $OD_{600}$ of 0.25 and incubated for 3 hr at 30°C.~5 $OD_{600}$ cells were quenched with 60% methanol at −40°C, and metabolites were extracted. For $^{15}$N-label incorporation in amino acids and nucleotides, cells were grown overnight in rich media, washed once and subsequently sub-cultured in minimal media (0.67% yeast nitrogen base without amino acids and ammonium sulfate, 0.1% glucose, 20 mM ammonium sulfate) at an initial $OD_{600}$ of 0.1 and grown till the $OD_{600}$ reached 0.5. At this point, $^{15}$N$_2$-ammonium sulfate (299286, Sigma-Aldrich) was added to reach a ratio of 50% unlabeled to 50% fully labelled ammonium sulfate. Metabolites were extracted from ~6 $OD_{600}$ cells. For $^{13}$C- label incorporation in nucleotides and other central carbon metabolites, cells were grown overnight in rich media, washed once and subsequently sub-cultured in minimal media at an initial $OD_{600}$ of 0.1 and grown till the $OD_{600}$ reached 0.5. For experiments where $^{13}$C- label incorporation into nucleotides was measured, in medium supplemented with or without additional methionine, cells were grown overnight in rich media, and subsequently sub-cultured in fresh rich media at an initial $OD_{600}$ of 0.2 and grown till the $OD_{600}$ reached 1, washed once and shifted to minimal media, 2% glucose with or without 2 mM methionine for 1 hr. After this time, [U-$^{13}$C$_6$] glucose (CLM-1396-PK, Cambridge Isotope Laboratories) was added to reach a ratio of 50% unlabeled to 50% fully labelled glucose. Metabolites were extracted from ~6 $OD_{600}$ cells. Extensive metabolite extraction protocols are described (*Walvekar et al., 2018a*). Metabolites were analyzed using LC-MS/MS method as described in *Walvekar et al. (2018a)*. Standards were used for developing multiple reaction monitoring (MRM) methods on Thermo Scientific TSQ Vantage Triple Stage Quadrupole Mass Spectrometer or Sciex QTRAP 6500. All the parent/product masses relevant to this study are listed in *Supplementary file 3*. Amino acids were detected in the positive polarity mode. For nucleotide measurements, nitrogen base release was monitored in the positive polarity mode. Trehalose was detected in the negative polarity mode. For PPP metabolites and other triose phosphates, phosphate release was monitored in the negative polarity mode.

Metabolites were separated using a Synergi 4µ Fusion-RP 80A column (100 × 4.6 mm, Phenomenex) on Agilent's 1290 infinity series UHPLC system coupled to the mass spectrometer. For positive polarity mode, buffers used for separation were- buffer A: 99.9% H$_2$O/0.1% formic acid and buffer B: 99.9% methanol/0.1% formic acid (Column temperature, 40°C; Flow rate, 0.4 ml/min; T = 0 min, 0% B; T = 3 min, 5% B; T = 10 min, 60% B; T = 11 min, 95% B; T = 14 min, 95% B; T = 15 min, 5% B; T = 16 min, 0% B; T = 21 min, stop). For ADP and ATP separation and detection, buffers used for separation were- buffer A: 5 mM ammonium acetate in H$_2$O and buffer B: 5 mM ammonium acetate in 100% methanol, and metabolites were measured in positive polarity mode. Alternately, buffers used for separation were- buffer A: 5 mM ammonium acetate in H$_2$O and buffer B: 100% acetonitrile (Column temperature, 25°C; Flow rate: 0.4 ml/min; T = 0 min, 0% B; T = 3 min, 5% B; T = 10 min, 60% B; T = 11 min, 95% B; T = 14 min, 95% B; T = 15 min, 5% B; T = 16 min, 0% B; T = 21 min, stop), and negative polarity mode was used. The area under each peak was calculated using Thermo Xcalibur software (Qual and Quan browsers) and AB SCIEX MultiQuant software 3.0.1.

## Spotting assay for comparative cell growth estimation

For all spotting assays, cells were grown overnight in rich media, washed once and subsequently sub-cultured in minimal media at an initial $OD_{600}$ of 0.2–0.25 and grown till the $OD_{600}$ reached 0.8–1.0. Cells were harvested by centrifugation, washed once with water and 10 µl sample for each suspension was spotted in serial 10-fold dilutions. For 8-aza adenine and hydroxyurea sensitivity assays,

cells were spotted onto minimal media plates containing 250 and 300 µg/ml 8-aza adenine (A0552, TCI chemicals) or 150 mM hydroxyurea (H8627, Sigma-Aldrich) and incubated at 30°C. For control plates without drug, images were taken after 1–2 days and for drug containing plates after 4–5 days. For genetic interaction analysis with Tps2, cells were spotted onto high and low Pi media plates with 2% glucose. Plates were incubated at 30°C and 37°C. For genetic interaction analysis with Pho80, cells were spotted onto minimal media plates with 0.1% and 2% glucose. Plates were incubated at 30°C.

## Trehalose and glycogen measurements

For trehalose and glycogen measurements in wild-type and thiolation mutants, cells were grown overnight in rich media, washed once and subsequently sub-cultured in minimal media at an initial $OD_{600}$ of 0.1 and grown till the $OD_{600}$ reached 0.8–1.0. For trehalose measurement in high and no Pi media, cells were grown overnight in rich media, washed once and subsequently sub-cultured in either high or no Pi media at an initial $OD_{600}$ of 0.1 and incubated for 8 hr at 30°C. For trehalose measurement in sulfur-rich and sulfur-starved conditions, wild-type cells were grown overnight in rich media, washed once and subsequently sub-cultured in sulfur-rich and sulfur-starved conditions at an initial $OD_{600}$ of 0.25 and incubated for 5 hr at 30°C. Cells were harvested by centrifugation and washed with ice-cold water. Cells were lysed in 0.25 M sodium carbonate by incubating at 95–98°C for 4 hr. Subsequently, added 0.15 ml 1M acetic acid and 0.6 ml of 0.2 M sodium acetate to bring the solution to pH 5.2. Trehalose and glycogen were digested overnight using trehalase (T8778, Sigma-Aldrich) and amyloglucosidase (10115, Sigma-Aldrich) respectively. Glucose released from these digestions was measured using a Glucose (GO) Assay Kit (GAGO20, Sigma-Aldrich). The concentration of released glucose (µg/ml) was determined from the standard curve and plotted. Statistical significance was determined using Student $T$-test (GraphPad Prism 7).

## Continuous chemostat culture growth to study yeast metabolic cycles, and microscopic analysis

Continuous chemostat cultures to establish the YMC were performed as described previously (*Tu et al., 2005*). An overnight batch culture of prototrophic CEN.PK strain (*van Dijken JP et al., 2000*) grown in rich medium was used to inoculate working volume of 1L in the chemostat. At 20 min time-intervals, cells were fixed with 2% paraformaldehyde, and imaged under a bright-field microscope. ~200 cells from each time point were sampled, and budding cells were counted manually.

Cell cycle synchronization and flow cytometry analysis bar1Δ::Hyg, uba4Δ::NAT bar1Δ::Hyg and ncs2Δ::NAT bar1Δ::Hyg cells were grown overnight in minimal media and subsequently sub-cultured in minimal media at an initial $OD_{600}$ of 0.05 and grown till the $OD_{600}$ reached 0.2. Cells were harvested by centrifugation, washed with water and resuspended in the same medium containing 10 µg/ml of α-factor (GenScript). Cells were kept at 30°C for 3 hr till complete G1 arrest was observed by light microscopy. Subsequently, 5 ml culture was harvested by centrifugation, washed with water and fixed with 70% ethanol for G1-arrested population. Remaining culture was synchronously released into the cell cycle by washing away the α-factor. Cells were collected at different intervals of time post G1 release, fixed with ethanol, treated with RNaseA (R4875, Sigma-Aldrich) and a protease solution (P6887, Sigma-Aldrich) as described (Haase and Reed, Cell cycle, 2002). Cells were stained with SYTOX green (S7020, Invitrogen) and analyzed on BD FACS Verse flow cytometer.

## Time-lapse live cell microscopy

Cells were grown overnight in rich media, washed once and subsequently sub-cultured in minimal media at an initial $OD_{600}$ of 0.1 and grown till the $OD_{600}$ reached 0.4–0.5. 1.5% agar pads (50081, Lonza) were prepared containing minimal media. The pad was cut into small pieces after it solidified. 2 µl of the cell suspension was placed on the agar pad, which was inverted and placed in a glass bottom confocal dish (101350, SPL Life Sciences) for imaging. Phase-contrast images were captured after every 3 min' interval for total of 360 mins on ECLIPSE Ti2 inverted microscope (NIKON) and 60X oil-immersion objective. Images were stacked and analyzed using ImageJ software. Statistical significance was determined using a Student $T$-test (GraphPad Prism 7).

## Transcriptome and ribosome profiling analyses

Cells were grown overnight in rich media and subsequently shifted to minimal media till the $OD_{600}$ reached 0.5–0.8. Cells were rapidly harvested by filtration and lysed, as described in detail (*Weinberg et al., 2016*; *McGlincy and Ingolia, 2017*). For both transcriptome and ribosome profiling analyses, three biological replicates each for WT and tRNA thiolation mutant cells (*uba4Δ* and *ncs2Δ*) were included. Total RNA and ribosome-protected fragments were isolated from the cell lysates and RNA-seq and ribosome profiling were performed, as described (*Weinberg et al., 2016*; *McGlincy and Ingolia, 2017*), with minor modifications. Separate 5' and 3' linkers were ligated to the RNA- fragment instead of 3' linker followed by circularization (*Subtelny et al., 2014*). 5' linkers contained four random nt unique molecular identifier (UMI) similar to a five nt UMI in 3' linkers. During size-selection, we restricted the footprint lengths to 18–34 nts. Matched RNA-seq libraries were prepared using RNA that was randomly fragmentation by incubating for 14 min at $95^0$C with in 1 mM EDTA, 6 mM $Na_2CO_3$, 44 mM $NaHCO_3$, pH 9.3. RNA-seq fragments were restricted to 18–50 nts. Ribosomal rRNA were removed from pooled RNA-seq and footprinting samples using RiboZero (Epicentre MRZH116). cDNA for the pooled libraries were PCR amplified for 16 cycles.

## Ribosome profiling data processing and analysis

RNA-seq and footprinting reads were mapped to the yeast transcriptome using the riboviz pipeline (*Carja et al., 2017*). Sequencing adapters were trimmed from reads using Cutadapt 1.14 (*Martin, 2011*) (–trim-n -e 0.2 –minimum-length 24). The reads from different samples were separated based on the barcodes in their 3' linkers using fastx_barcode_splitter (FASTX toolkit, Hannon lab) with utmost one mismatch allowed. UMI and barcodes were removed from reads in each sample using Cutadapt (–trim-n -m 10 u 4 u −10). Trimmed reads that aligned to yeast rRNAs and tRNAs were removed using HISAT2 v2.1.0 (*Kim et al., 2015*). Remaining reads were mapped to a set of 5812 genes in the yeast genome (SGD version R64-2-1_20150113) using HISAT2. Only reads that mapped uniquely were used for all downstream analyses. Codes for generating processed fastq and gff files were obtained from riboviz package (https://github.com/shahpr/RiboViz; *Carja et al., 2017*). Gene-specific fold-changes in RNA and footprint abundances were estimated using DESeq2 packages in R (*Love et al., 2014*) using default log-fold-change shrinkage options. Changes in ribosome-densities (translation efficiencies) were estimated using the Riborex package in R (*Li et al., 2017*).

The complete transcript/ribosome footprint datasets are available at GEO (number GSE124428; link: https://www.ncbi.nlm.nih.gov/geo/query/acc.cgi?acc=GSE124428).

## Acid phosphatase assay for Pho5 and related enzyme activity

Cells were grown overnight in rich media, washed once and subsequently sub-cultured in high and low Pi media (for experiment related to wild-type and *uba4Δ*) or complete media (for experiment related to wild-type, *uba4Δ*, *gcn4Δ* and *uba4Δ gcn4Δ*) at an initial $OD_{600}$ of 0.1 and grown till the $OD_{600}$ reached 0.5–0.6. 8 $OD_{600}$ cells were collected, washed once and resuspended in sterile water to final $OD_{600}$ of 16. Acid phosphatase activity was assayed, as described previously with some modifications (*Huang and O'Shea, 2005*). Briefly, 450 μl of each cell suspension was added to 200 μl of 20 mM p-nitrophenyl phosphate (PNPP, N4645, Sigma-Aldrich) in 0.1M sodium acetate, pH-4.2, mixed and incubated at room temperature for 30 min. To stop the reaction, 200 μl of the reaction mixture was withdrawn and added to 200 μl of ice cold 10% trichloroacetic acid. To this reaction, 400 μl of saturated sodium carbonate solution (2M, pH-11.5) was added, mixed and centrifuged at 3000 rpm for 10 min. 80 μl of the supernatant was transferred in technical triplicates to a 96-well plate and liberated p-nitrophenol was determined by measuring $OD_{420}$ on a plate reader. Phosphatase activity was measured in units expressed as $OD_{420}/OD_{600} \times 1000$. Statistical significance was determined using a Student T-test (GraphPad Prism 7).

## Phosphate measurement

Cells were grown overnight in rich media, washed once and subsequently sub-cultured in minimal media at an initial $OD_{600}$ of 0.1 and grown till the $OD_{600}$ reached 0.8–1.0. Free intracellular phosphate levels were determined, as described previously (*McNaughton et al., 2010*). Briefly, cells were harvested by centrifugation and washed twice with ice-cold water. Cells were lysed by resuspending in 200 μl 0.1% triton X-100 and vortexed for 5 min with glass beads. The lysate was cleared

by centrifugation and protein concentration was determined by using bicinchoninic acid assay (23225, Thermo Fisher). 30 µg of whole cell lysate was used for measurement of free intracellular phosphate levels using ammonium molybdate and ascorbic acid colorimetric assay as described (*Ames, 1966*). Potassium dihydrogen phosphate solution was used for standard curve (0 to 500 µM $KH_2PO_4$). The amount of phosphate was expressed as µM Pi. Statistical significance was determined using a Student *T*-test (GraphPad Prism 7).

## Luciferase assay for *GCN4* translation

WT, uORF1* and uORF4* Gcn4-luciferase reporter plasmids (SL148, SL149 and SL150) were generated by PCR amplification of a 777 bp fragment of WT, uORF1*and uORF4* Gcn4 constructs (GeneArt, Thermo Scientific) and subsequent cloning in SL147 plasmid (*Supplementary file 2*). SL147 was generated by cloning luciferase cDNA from pGL3-Basic (Addgene) in pSL80 plasmid (*Wu and Tu, 2011*). For luciferase assay in wild-type, thiolation mutants, *gcn2Δ* and double mutants, cells transformed with Gcn4-luciferase reporter plasmids were grown overnight in rich media in presence of G418. Cells were subsequently sub-cultured in the same media without selection till logarithmic phase and then shifted to minimal media. After 4–5 hr of growth ($OD_{600}$ of 0.8–1.0), cells were collected, washed twice with ice-cold lysis buffer (1X PBS, 1 mM PMSF). For luciferase assay in wild-type and *gcn2Δ*, in high and no Pi media, cells transformed with Gcn4-luciferase reporter plasmid were grown overnight in rich media in presence of G418. Cells were subsequently sub-cultured in high and no Pi media supplemented with amino acids, 2% glucose and incubated for 8 hr at 30˚C, cells were collected, washed twice with ice-cold lysis buffer (1X PBS, 1 mM PMSF). For luciferase assay in sulfur amino acid limited condition, wild-type cells transformed with Gcn4-luciferase reporter plasmid were grown overnight in rich media, and subsequently sub-cultured in rich media at an initial $OD_{600}$ of 0.2 and grown till the $OD_{600}$ reached 0.4–0.5 and half of the cells were collected for luciferase assay. Remaining cells were washed once and subsequently shifted to sulfur amino acid limited condition and incubated for 1 hr at 30˚C. Lysis was done by vortexing cells for 5 min in presence of glass beads. Lysates were cleared by centrifugation and protein concentration was determined by using a bicinchoninic acid assay (23225, Thermo Fisher). Luciferase assay was performed using luciferase assay system kit (E1500, Promega) and activity was measured using a Sirius luminometer (Tiertek Berthold). Data output provided as Relative Light Units per sec (RLU/s) was used to determine relative luciferase activity. Statistical significance was determined using a Student *T*-test (GraphPad Prism 7).

## RNA isolation and quantitative real-time PCR (qRT-PCR) analysis

Cells were grown overnight in rich media, washed once and subsequently sub-cultured in minimal media at an initial $OD_{600}$ of 0.1 and grown till the $OD_{600}$ reached 0.8–1.0. Cells were harvested by centrifugation, and RNA was isolated by hot phenol beating method (*Collart and Oliviero, 2001*. 6 µg of total RNA was used for DNase I (AM2238, Thermo Fisher) treatment. cDNA was synthesized with random primers (48190011, Thermo Fisher) and SuperScript II reverse transcriptase (18064014, Thermo Fisher Scientific) according to the manufacturer's protocol. cDNA quantification was done by real-time PCR on an ViiA 7 Real-Time PCR System (Thermo Fisher) using Maxima SYBR Green/ ROX qPCR Master Mix (K0222, Thermo Fisher). ACT1 was used as an internal normalization control. All qRT-PCRs were performed in triplicates using at least two independent biological RNA samples. Statistical significance was determined using a Student *T*-test (GraphPad Prism 7).

## ATP measurement

Total cellular ATP was measured using the ATP determination kit (A22066, Thermo Fisher) according to the manufacturer's protocol. Briefly, 5 µl sample or 5 µl of different concentrations of ATP standard solution (0 to 5 µM ATP) was added to 50 µl of assay solution. Luminescence was measured directly after addition of the sample to assay solution using a Sirius luminometer (Tiertek Berthold). ATP concentrations in samples were calculated from the ATP standard curve and relative levels were plotted. Statistical significance was determined using a Student *T*-test (GraphPad Prism 7).

## Acknowledgements

We acknowledge the use of the NCBS/inStem/CCAMP mass spectrometry facility for LC-MS/MS instrument support. We thank Dr. Anjana Badrinarayanan for the use of her live-cell imaging microscope system. We thank Claudio de Virgilio, Nikolai Slavov, Sider Penkov, and Sriram Varahan for critical comments on this manuscript. RG and ASW acknowledge the Department of Science and Technology and Science and Engineering Research Board (DST-SERB) national postdoctoral fellowships (PDF/2016/000416 and PDF/2015/000225 respectively). SLA is supported by an Intermediate Fellowship from the Wellcome Trust-DBT India Alliance (grant number IA/I/14/2/501523), as well as institutional support from inStem and the Dept. of Biotechnology (Govt. of India). PS is supported by grants NIH R35 GM124976, and subcontracts from NIH R01 DK056645 and NIH R01 DK109714 as well as start-up funds from the Human Genetics Institute of New Jersey at Rutgers University.

## Additional information

### Funding

| Funder | Grant reference number | Author |
| --- | --- | --- |
| Science and Engineering Research Board, DST | PDF/2016/000416 | Ritu Gupta |
| Science and Engineering Research Board, DST | PDF/2015/000225 | Adhish S Walvekar |
| Wellcome Trust/DBT India Alliance | IA/I/14/2/501523 | Sunil Laxman |
| Department of Biotechnology, Govt. of India | | Sunil Laxman |
| inStem | | Sunil Laxman |
| National Institutes of Health | R35 GM124976 | Premal Shah |
| National Institutes of Health | R01 DK056645 | Premal Shah |
| National Institutes of Health | R01 DK109714 | Premal Shah |
| Human Genetics Institute of New Jersey at Rutgers University | Start-up funds | Premal Shah |

The funders had no role in study design, data collection and interpretation, or the decision to submit the work for publication.

### Author contributions

Ritu Gupta, Data curation, Formal analysis, Validation, Investigation, Methodology, Writing—original draft, Writing—review and editing; Adhish S Walvekar, Data curation, Formal analysis, Validation, Methodology; Shun Liang, Data curation, Validation; Zeenat Rashida, Data curation, Formal analysis, Validation; Premal Shah, Conceptualization, Resources, Software, Formal analysis, Supervision, Funding acquisition, Visualization, Writing—review and editing; Sunil Laxman, Conceptualization, Resources, Data curation, Formal analysis, Supervision, Funding acquisition, Visualization, Writing—original draft, Writing—review and editing

### Author ORCIDs

Ritu Gupta https://orcid.org/0000-0002-0563-6599
Adhish S Walvekar https://orcid.org/0000-0001-7344-7653
Premal Shah https://orcid.org/0000-0002-8424-4218
Sunil Laxman https://orcid.org/0000-0002-0861-5080

### Decision letter and Author response

Decision letter https://doi.org/10.7554/eLife.44795.030
Author response https://doi.org/10.7554/eLife.44795.031

# Additional files

### Supplementary files

• Supplementary file 1. Strains used in this study.
DOI: https://doi.org/10.7554/eLife.44795.022

• Supplementary file 2. Plasmids used in this study.
DOI: https://doi.org/10.7554/eLife.44795.023

• Supplementary file 3. Mass transitions for detection of metabolites.
DOI: https://doi.org/10.7554/eLife.44795.024

• Supplementary file 4. RNA-RFP changes (mutant-WT).
DOI: https://doi.org/10.7554/eLife.44795.025

• Transparent reporting form
DOI: https://doi.org/10.7554/eLife.44795.026

### Data availability

Sequencing (transcript and ribosome footprint) data have been deposited in GEO under accession codes GSE124428, and are fully open.

The following dataset was generated:

| Author(s) | Year | Dataset title | Dataset URL | Database and Identifier |
|---|---|---|---|---|
| Gupta R, Walvekar AS, Liang S, Rashida Z, Shah P, Laxman S | 2019 | A tRNA modification balances carbon and nitrogen metabolism by regulating phosphate homeostasis | https://www.ncbi.nlm.nih.gov/geo/query/acc.cgi?acc=GSE124428 | NCBI Gene Expression Omnibus, GSE124428 |

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
