## [Decision Letter]

Thank you for submitting your article "A tRNA modification balances carbon and nitrogen metabolism by regulating phosphate homeostasis" for consideration by *eLife*. Your article has been reviewed by two peer reviewers, including Alan Hinnebusch as the Reviewing Editor, and the evaluation has been overseen by Naama Barkai as the Senior Editor. The following individual involved in review of your submission has also agreed to reveal his identity: Bertrand Daignon-Fornier (Reviewer #2).

The reviewers have discussed the reviews with one another and the Reviewing Editor has drafted this decision to help you prepare a revised submission.

Summary:

This paper presents evidence that in yeast mutants incapable of thiolating tRNAs the expression of many genes of the PHO phosphate regulon is reduced, along with intracellular phosphate (Pi) levels. This apparently accounts for the decreased nucleotide synthesis and nucleotide abundance observed in these mutants, with attendant induction of Gcn2-dependent Gcn4 translation and amino acid biosynthesis (shown previously to be induced by purine limitation). The mutants also exhibit elevated trehalose, whose synthesis by Tps2 is shown to mitigate the reduction in intracellular Pi levels and to support cell viability in the thiolation mutants. Furthermore, Pi starvation was shown to induce trehalose and Gcn4 in wild-type cells (although nucleotide synthesis/levels were not measured and probably should have been). They propose a model that tRNA thiolation is an intracellular signal for methionine/cysteine availability and that when thiolation drops it reduces nucleotide synthesis and cell division as an adaptive response by down-regulating PHO gene expression and intracellular Pi levels. This in turn diverts glucose utilization from glycolysis to trehalose synthesis, liberating Pi from glucose-6-Pi to ameliorate the Pi starvation and slow down cell growth and metabolism. The study does not provide a molecular basis for the down-regulation of PHO gene transcripts in the thiolation mutants; nor was it shown that starvation for Met or Cys would elicit the key phenotypic responses observed in the thiolation mutants, which would seem to be essential evidence for their model if it is not already demonstrated in the literature. The study also provides an apparently overlooked explanation for why trehalose increases on phosphate starvation-to release Pi from Glu-6-Pi-in addition to diverting glucose to a storage carbohydrate like glycogen; although they seem to overlook the fact that increased trehalose levels is a common response to many different kinds sorts of cellular stress.

Essential revisions:

1) Perform new experiments to show that limitation for the sulfur-containing amino acids and/or inorganic sulfate, elicits the same key responses seen for the thiolation mutants, to support the claim that tRNA thiolation is used as a sensor for sulfur levels.

2) Cite the appropriate literature from the Rabinowitz and Broach labs (Boer et al., 2010 and Klosinslka et al., 2011) showing that nucleotide levels are reduced on Pi starvation, to support the claim that the reduced nucleotide levels in the thiolation mutants can be attributed to the reduced Pi levels in these strains.

3) Measure the ATP/GTP levels (or at least AXP/GXP) in the thiolation mutants rather than relying on measurements of the much less abundant AMP/GMP.

4) Expand the analysis of the RNA-Seq data of Figure 5 to show and discuss all of the transcriptional changes for genes in the PPP and trehalose and glycogen pathways.

5) Use the proper time frame for the experiment in Figure 2 in order to measure the flux in the PPP and glycolysis pathways in the thiolation mutants, to confirm the rerouting hypothesis.

6) Discuss the possibility that lower flux in the PPP might result in low NADPH and attendant diminished sulfate assimilation under conditions where sulfur is already limiting. Reduced sulfate assimilation could in turn limit thiolation and amplify the whole regulatory response.

*Reviewer #1:*

This paper presents evidence that in yeast mutants incapable of thiolating tRNAs the expression of many genes of the PHO phosphate regulon is reduced, along with intracellular phosphate (Pi) levels. This apparently accounts for the decreased nucleotide synthesis and nucleotide abundance observed in these mutants, with attendant induction of Gcn2-dependent Gcn4 translation and amino acid biosynthesis (shown previously to be induced by purine limitation). The mutants also exhibit elevated trehalose, whose synthesis by Tps2 is shown to mitigate the reduction in intracellular Pi levels and to support cell viability in the thiolation mutants. Furthermore, Pi starvation was shown to induce trehalose and Gcn4 in wild-type cells (although nucleotide synthesis/levels were not measured and probably should have been). They propose a model that tRNA thiolation is an intracellular signal for methionine/cysteine availability and that when thiolation drops it reduces nucleotide synthesis and cell division as an adaptive response by down-regulating PHO gene expression and intracellular Pi levels. This in turn diverts glucose utilization from glycolysis to trehalose synthesis, liberating Pi from glucose-6-Pi to ameliorate the Pi starvation and slow down cell growth and metabolism. The study does not provide a molecular basis for the down-regulation of PHO gene transcripts in the thiolation mutants; nor was it shown that starvation for Met or Cys would elicit the key phenotypic responses observed in the thiolation mutants, which would seem to be essential evidence for their model if it is not already demonstrated in the literature. The study also provides an apparently overlooked explanation for why trehalose increases on phosphate starvation-to release Pi from Glu-6-Pi-in addition to diverting glucose to a storage carbohydrate like glycogen; although they seem to overlook the fact that increased trehalose levels is a common response to many different kinds sorts of cellular stress.

- It seems crucial to determine whether starvation for Met or Cys elicits decreased expression of the PHO regulon, decreased Pi levels (in the uba4 tps2 double mutant where it can most easily be measured), and increased trehalose levels in order to provide key support for their model that tRNA thiolyation is the intracellular sensor for Met/Cys levels and used to re-wire phosphate, nucleotide, and trehalose metabolism in the cell.

- Figure 1—figure supplement 1A: It's also unexpected that the mutations would increase expression from the uORF1* and uORF4* mutant reporters. They should examine the reporter lacking both uORF1 and uORF4 to determine whether the increased expression in the mutants is eliminated by removing both uORFs.

- Can the S phase delay, as well as the induction of Gcn4, in the thiolation mutants be rescued by supplementing cultures with nucleoside precursors or ribose-5-P to confirm that these phenotypes result from low nucleotide levels?

- Subsection “Phosphate depletion in wild-type cells phenocopies tRNA thiolation mutants” and Figure 5D: It should be determined whether the increase in Gcn4 protein level elicted by Pi starvation is abolished in a gcn2 mutant, as would be expected if phosphate starvation is mimicking the thiolation mutants.

- It seems important to determine whether nucleotide levels are reduced during Pi starvation, unless shown in the literature already, to support the claim that the reduced nucleotide levels in the thiolation mutants result from their reduced Pi levels.

- Subsection “Trehalose synthesis associated phosphate release enables cells to maintain phosphate balance”, second paragraph and Figure 6D: PHO gene expression is reduced in the thiolyation mutants. Why then should constitutively activating PHO genes under the control of Po2/Pho4 in a pho80 mutant be synthetic-sick with the thiolyation mutants? This needs to be explained more precisely than just a loss of phosphate homeostasis, which is too vague.

- Given that increased trehalose synthesis is a common response to different kinds of stress e.g. trehalose is an osmoprotectant – isn't it dangerous to ascribe to increased trehalose synthesis a specific role in mitigating the effects of reduced Pi levels in the thiolation mutants, and by extension, conditions of Met/Cys limitation? Wouldn't it make more sense to emphasize the role of increased trehalose synthesis in depleting Glucose-6-P levels, with attendant reduction in nucleotide synthesis and cell division when cells are limited for Met/Cys?

*Reviewer #2:*

The manuscript entitled « A tRNA modification balances carbon and nitrogen metabolism by regulating phosphate homeostasis. » by Gupta and co-workers reveals a very interesting crosstalk between metabolic pathways in yeast. The authors take advantage of mutants lacking a specific tRNA-modification to reveal and decipher complex interactions between metabolic pathways. This work hence nicely illustrates how cells integrate signals resulting from various nutrient inputs. The general scheme emerging from the work is that lack of tRNA-thiolation somehow affects the regulation of phosphate metabolism and results in lower intracellular inorganic-phosphate. As a consequence, glucose-6P would be rerouted from the pentose phosphate pathway (PPP) to trehalose synthesis which releases inorganic phosphate. Lower flux in the PPP would limit nucleotide synthesis and thereby activate a Gcn4-dependent response.

Each of the metabolic and expression steps is well documented, however most of the mechanisms connecting the regulation processes are not elucidated nor discussed. Addressing (when possible) and/or discussing the following questions would improve and strengthen the manuscript:

1) What is the physiological signal mimicked by thiolation defective mutants? Since tRNA-thiolation reflects intracellular availability of sulfur containing amino-acids (Introduction, fourth paragraph) and since sulfur is mostly used for synthesis of methionine, cysteine and S-adenosylmethionine (Thomas and Kerjan, 1997, p. 503-532), does thiolation mirror sulfur availability? Would sulfur limitation recapitulate the thiolation-defect? this would strongly increase the overall interest of the paper.

2) What is the connection between tRNA-thiolation and PHO-genes down-regulation? Simple experiments would allow to genetically position the tRNA-thiolation mutants within the phosphate regulation scheme. For example, do the ncs2/uba4 mutants affect Pho4 nuclear localization? Systematic additivity and epistasis studies with PHO regulation-mutants could be carried out. Unexpectedly the double ncs2/uba4 mutants with pho80 (subsection “Trehalose synthesis associated phosphate release enables cells to maintain phosphate balance”, second paragraph) reveal a synthetic growth phenotype while these mutants should antagonize each other. As such this experiment does not support the model.

3) It is not clear how the rerouting from the PPP to trehalose synthesis takes place. The authors should show (Figure 4) and discuss the transcriptional response for all the genes of the PPP as well as the trehalose and glycogen pathways. Is transcriptional regulation responsible for the rerouting? Importantly the chosen time frame (Figure 2 and subsection “Carbon flux is routed towards storage carbohydrates in thiolation mutants”, first paragraph) did not allow to directly measure the flux in the PPP and glycolysis and therefore the rerouting hypothesis relies only on trehalose, AMP and GMP synthesis.

4) The proposed PPP to trehalose rerouting allows to recycle inorganic phosphate when carbon is plentiful and phosphate is scarce, however the lower flux in the PPP might result in low NADPH which is crucial for sulfate assimilation (the zwf1/met19 mutant is a strict methionine-auxotroph). Hence a decreased flux in the PPP could limit sulfur assimilation under conditions where sulfur might already be limiting (see point 1 above).

5) In the proposed model, activation of the Gcn4-dependent pathway is merely a consequence of the rerouting and is not responsible for lower phosphate or higher trehalose. This assumption could easily be verified by combining ncs2/uba4 and gcn4 mutations and measuring phosphate and trehalose in the double mutants.

6) The effect of tRNA-thiolation on nucleotides synthesis should also be documented at the level of tri-phosphates (not only AMP and GMP) since ATP and GTP are by far the most abundant nucleotides and more directly reflect intracellular nucleotide content. At least AXP and GXP could be shown. Similarly, measurements of intracellular dNTPs would be relevant to confirm cell cycle analyses/hypotheses.

7) I do not understand how the authors can normalize on "cell number as well as biomass". I could not find the information in the Materials and methods section nor in the figure legend. This is a very important point that should be clarified. Cell number imperfectly reflects biomass if not corrected by the cell volume.

[Editors' note: further revisions were requested prior to acceptance, as described below.]

Thank you for resubmitting your work entitled "A tRNA modification balances carbon and nitrogen metabolism by regulating phosphate homeostasis" for further consideration at *eLife*. Your revised article has been favorably evaluated by Naama Barkai as the Senior Editor, and two reviewers, one of whom is a member of our Board of Reviewing Editors.

The manuscript has been improved but there are some remaining issues that need to be addressed before acceptance, as outlined in each of the two reviews shown below. Please make the appropriate revisions and provide a brief point-by-point explanation in response to the remaining reviewers' comments.

*Reviewer #1:*

I am generally satisfied that the addition of new experimental data and revisions of text address the major concerns with the previous version of the paper, except that the GCN4 mRNA reads still were not added to Figure 4—figure supplement 2.

*Reviewer #2:*

(Partially rephrased by Deputy Editor Detlef Weigel for clarity.)

General comment: the essential response points by the authors are difficult to extract from pages of justification which in my opinion turbidify rather than clarify the issues. Several of the original concerns have been satisfactorily addressed, but the following remain:

1) Perform new experiments to show that limitation for the sulfur-containing amino acids and/or inorganic sulfate, elicits the same key responses seen for the thiolation mutants, to support the claim that tRNA thiolation is used as a sensor for sulfur levels.

Figure 2G shows that absence of methionine elicits a much stronger response than the thiolation mutant does. Indeed, the increased flux toward AMP in the presence of methionine is diminished in the uba4 mutant but stays much higher than in the absence of methionine (same for either WT or uba4) and similar partial effects were observed for GMP (Figure 2—figure supplement 1D). These results indicate that tRNA thiolation contributes (very) partially to the methionine (sulfur) signal (when purine synthesis is used as a readout) suggesting the existence of other (yet unknown) mechanisms connecting sulfur and carbon metabolism. This should be discussed.

4) Expand the analysis of the RNA-Seq data of Figure 5 to show and discuss all of the transcriptional changes for genes in the PPP and trehalose and glycogen pathways.

As now shown (Figure 4C and Figure 4—figure supplement 3C) and stated by the authors "there are very minor changes in either transcript or RPF for any of the genes in these pathways". Consequently, the molecular mechanism resulting in rerouting the carbon flux away from the PPP toward trehalose synthesis is clearly not at the RNA or protein expression levels and is not yet elucidated. Hence the last sentence of the Discussion "we biochemically explain how phosphate homeostasis determines flux through different arms of carbon and nitrogen metabolism" appears overstated to me.

---

## [Author Response]

Essential revisions:1) Perform new experiments to show that limitation for the sulfur-containing amino acids and/or inorganic sulfate, elicits the same key responses seen for the thiolation mutants, to support the claim that tRNA thiolation is used as a sensor for sulfur levels.

This is an important point that requires a substantial response, which also allows us to more clearly explain inferences made from earlier studies. Multiple lines of evidence, based on published work, along with this study, suggest that tRNA thiolation is not a starvation response, but a growth response, i.e. tRNA thiolation enables cells to fully respond to the *presence* of sulfur amino acids, and not starvation /absence of sulfur/sulfur amino acids. The loss of tRNA thiolation results in a more subtle metabolic phenotype, and understanding this is the primary point of this manuscript. To clarify further, the connection of tRNA thiolation (and related modifications) with sulfur amino acids and sulfur comes from our earlier work that demonstrate the following: (i) when comparing the amounts of thiolated tRNAs in methionine rich medium vs. medium with less methionine, the amount of thiolated tRNAs increased with increasing methionine availability and vice versa (see Laxman et al., 2013, Figures 1C, 1D, 1F, and supplementary figures). (ii) This same study quantified the absolute amounts of sulfur amino acid metabolites, and compared it with amounts of thiolated tRNAs. Both methionine and thiolated tRNAs (and closely related modified tRNAs) were present at low μM amounts in cells in normal medium, i.e. the amount of tRNA thiolation reflected the availability of methionine. (iii) The amounts of thiolated tRNAs are highest in cells entering a ‘growth state’ (high ribosomal biosynthesis, entry into the cell cycle etc.), as suggested from that study, as well as earlier gene expression studies (e.g. Tu et al., 2005). (iv) in thiolation mutants, the proteins involved in methionine salvage and biosynthesis increase in amounts over time (Laxman et al., 2013), further suggesting a coupling between the thiolation pathway and methionine. (v) the key enzyme in the thiolation pathway, Uba4, is strongly regulated by methionine availability, decreasing in methionine limited medium. These together show the coupling of tRNA thiolation with methionine/sulfur amino acids, pointing towards a role for tRNA thiolation when sulfur amino acids are abundantly present. Separately, several recent studies have suggested a multi-component growth program when methionine is abundant (including Sutter et al. Cell 2013, Yi et al. Mol Cell 2017). Pertinently, our recent study defined a methionine induced anabolic program, which centred around two arms: an increase in carbon metabolism (particularly PPP flux) towards nucleotide synthesis, and an increased utilization of amino acids for the same, when methionine is abundant (Walvekar et al., 2018). Therefore, to demonstrate this coupling of tRNA thiolation with the presence of sulfur amino acids, and the full response of sulfur amino acids, the more appropriate experiment will be to compare carbon flux towards nucleotides (via the PPP, as also elaborated in response to point #5 below) when abundant methionine is provided. Here, the prediction would be that if tRNA thiolation aided the methionine mediated metabolic program, this increase in nucleotide synthesis due to methionine would be less in thiolation mutants, compared to WT cells. We therefore carried out the following experiment: in WT cells growing in minimal glucose medium, or in this medium supplemented with methionine, using labeled 13C glucose pulsed into the system, we measured flux of carbon incorporation into nucleotides (also see response to point #5 below). Here, methionine strongly increases carbon flux into nucleotides in WT cells. Notably, this methionine dependent, increased flux through this pathway is strikingly lower in the thiolation mutants. This shows that for the full metabolic response due to methionine, leading to increased carbon flux towards nucleotide synthesis, tRNA thiolation is required. These data are now included in the new Figures 2G, Figure 2—figure supplement 1D), and are explained in detail in a new Results subsection (“Methionine induced carbon flux to nucleotide synthesis is dampened in thiolation mutants”), and Discussion. This data therefore more directly substantiates the coupling of tRNA thiolation with *increased* sulfur amino acid (methionine) availability, and in regulating the overall metabolic output mediated by methionine.

We note that sulfur starvation is a separate, far more complex phenomenon. Given the central nature of sulfur for a variety of essential biochemical functions, cells have a complex, coordinated response to this. This includes a downregulation of all methylation reactions due to reduced SAM, a strong induction of the MET regulon, and sulfur salvage/assimilation pathways, and a large reduction in growth (including low carbon metabolism), as has been demonstrated in different studies. When there is sulfur starvation, there is almost no thiolated tRNAs, or the enzyme required to make thiolated tRNAs, as has been demonstrated (Laxman et al., 2013). So a clear experiment in this condition to directly probe the coupling of tRNA thiolation with the *absence* sulfur amino acids, is not obvious, nor is it likely to clarify the question at hand.

We will separately also note that an earlier study (Boer VM et al., 2010) has made a connection between phosphate starvation (the experiment performed), and sulfur metabolism, where (we quote): “In phosphorus limitation, one of the most striking transcriptional patterns was, intriguingly, not directly related to phosphate metabolism, but instead to sulfate. SUL1, MMP1, MHT1, CYS3, MUP1, and SAM1 were strongly repressed by phosphorus limitation, with SUL1 in phosphorus limitation showing the most positive growth rate slope of any gene in any condition. This suggests a strong relationship between phosphate limitation and sulfur metabolism.”These data parallel our data (with sulfur metabolism as a starting point) where we note a coupling between sulfur metabolism and phosphate metabolism. If tRNA thiolation is indeed coupled with sulfur metabolism, then such a down-regulation of phosphate metabolism when tRNA thiolation is absent would therefore be a logical prediction.

Finally, in relation to this, and for the reference of the reviewers, we include some additional data. We and others have worked with conditions where reduced sulfur (methionine and related metabolites) are limiting (e.g. Xi and Tu BP MBoC 2011, Sutter et al. Cell 2013, Walvekar et al., 2018 etc.). We analyzed some transcription data sets from Walvekar et al., 2018, as well as some unpublished datasets (unrelated to this manuscript) from an ongoing study. Here, we find that in cells present in medium supplemented with methionine (compared to cells in somewhat methionine-limited conditions), several genes related to phosphate metabolism are induced. Here, in these condition phosphate is not limiting. While this correlates with observations made in this study, w.r.t the thiolation mutants, this also suggests a deeper coupling between sulfur and phosphate metabolism. This certainly deserves a more substantial investigation in future studies, well beyond the scope of this manuscript. We include this preliminary data in Author response image 1. We feel it is premature (and distracting) to include these data in this manuscript, since it takes away from the primary point of tRNA thiolation integrating methionine/sulfur sensing with overall metabolic outputs, via phosphate regulation. We hope collectively, we have sufficiently clarified and addressed all concerns raised.

2) Cite the appropriate literature from the Rabinowitz and Broach labs (Boer et al., 2010 and Klosinslka et al., 2011) showing that nucleotide levels are reduced on Pi starvation, to support the claim that the reduced nucleotide levels in the thiolation mutants can be attributed to the reduced Pi levels in these strains.

In our original submission, we had cited the Boer et al., 2010 and Klosinska et al., 2011 references, but had not fully described this important hallmark of Pi starvation, while describing how phosphate starvation phenocopied the metabolic switch observed in thiolation mutants. We have now done so, substantially clarifying the text corresponding to the section related to Figure 6, as well as in the Discussion section.

We also now include an additional citation (Saldanha et al., 2004). This study shows that phosphate starvation itself results in a decrease in ribosomal genes. In the thiolation mutants, we also see slightly reduced ribosomal genes (RNA and ribo-seq), as shown in Figure 4—figure supplement 2C. Thus, this is an additional converging datapoint, where phosphate starvation phenocopies tRNA thiolation mutants. This is added to the Results subsection (“Phosphate depletion in wild-type cells phenocopies tRNA thiolation mutants”).

Summarizing additions in the text; in the result section where we show that phosphate starvation in the WT cells phenocopies thiolation mutants, we state that reduced nucleotides are a feature of phosphate starvation in WT cells (citing Boer et al., 2010, Klosinska et al., 2011), and that reduced ATP is a signature feature of phosphate starvation. We summarize our presented data where we show reduced de novo nucleotide synthesis (using both nitrogen and carbon labeling), as well as reduced steady state nucleotides, with an emphasis on the reduced ATP amounts observed in the thiolation mutants. This is summarized in the aforementioned Results subsection, and in the revised Discussion section.

3) Measure the ATP/GTP levels (or at least AXP/GXP) in the thiolation mutants rather than relying on measurements of the much less abundant AMP/GMP.

This is an important point raised by the reviewers, therefore we would like to elaborate and clarify this point in detail. The reviewers are correct that at steady-state, the amounts of ATP/GTP are much higher than AMP/GMP. Indeed, nucleotides are synthesized as mono phosphates, and then eventually form di and triphosphates, and cycle between these states. Therefore, if measuring only steady state changes, it is essential to show levels of all these. However (in addition to steady-state levels of nucleoside monophosphates), we had included data from stable-isotope based flux analysis of nucleotide synthesis, using pulse-labels (of both nitrogen and carbon), using LC-MS/MS. Unlike changes observed in steady-state amounts of a metabolite (which can come due to decreased synthesis or increased utilization), this more directly addresses changes in rates of nucleotide synthesis, and since the di and tri phosphates are made from the monophosphates, presenting monophosphates alone is usually sufficient. Given this difference between steady-state measurements, and stable-isotope based flux experiments, it is now well accepted to show the synthesis of the nucleoside monophosphates, and to indicate carbon incorporation into nucleotides. This is commonly used in studies from cancer metabolism, where nucleotide synthesis is the focus (examples from the de Berardinis, Locasale, Vander Heiden and Maddocks groups: Huang F et al., Cell Metabolism 2018, 28(3):369-382.e5, Reid MA et al., Nature Communications 2018, 9: 5442, Lunt et al. Mol Cell 2015, Labuschagne et al. Cell Reports 2014). We have ourselves used this approach in other studies (e.g. descriptive methods paperWalvekar et al., 2018, Laxman et al. Sci Signal 2014, Walvekar et al., 2018 etc.). One reason for measuring NMPs alone; while using LC-mass spectrometry, resolving and detecting NXPs have different challenges. NMPs are easily detected in positive-polarity modes (MS/MS), and separated in aqueous-organic phases like 0.1% formic acid/methanol. NTPs however ionize and fragment best in negative polarity modes, and require different reverse phase columns/solvents to efficiently separate them. So it is difficult to get both NMP and NTP information within the same MS run.

However, since this was a particular point raised by the reviewers, we have carried out separate experiments and included ADP and ATP measurements. We compared steady-state ATP levels in WT and thiolation mutants using standard biochemical assays, and saw a similar reduction in steady state ATP as we did for steady state NMPs. This is included as a supporting figure (Figure 1—figure supplement 2A and B). Next, we developed/optimized a new LC-MS/MS method where AMP, ADP and ATP could all be reliably, quantitatively measured in the same run. This is now described in the Materials and methods section. Using this, we repeated a 13C-glucose stable-isotope based flux experiment (as we had done for NMPs), and measured relative label incorporation into ADP and ATP as well. Here, we clearly observed a significant decrease in the relative label incorporation into ADP and ATP in the thiolation mutants, largely mirrorring the data shown earlier for AMP and GMP. These data substantially strengthen the observation that the thiolation mutants have reduced nucleotide synthesis, and are now included as a new figure panel, now Figure 2C, D, and collectively summarized in the Results section primarily related to Figure 2.

4) Expand the analysis of the RNA-Seq data of Figure 5 to show and discuss all of the transcriptional changes for genes in the PPP and trehalose and glycogen pathways.

We thank the reviewers for this suggestion. We have included the analysis of transcriptional and ribosome-footprint changes in the genes related to the PPP, as well as trehalose/glycogen synthesis. This is now included as a new figure (Figure 4D), and also described in the text in the relevant Results subsection (“Loss of tRNA thiolation results in reduced phosphate homeostasis (PHO) related transcripts and ribosome-footprints”, third paragraph). As can be seen, there are very minor changes in either transcript or RPF for any of the genes in these pathways, in the thiolation mutants compared to WT cells. Indeed, including this data as a figure reiterates that the metabolic rewiring, in central carbon metabolism, is largely dictated by flux and mass-action. This strengthens our assertion that the causal control point in the tRNA thiolation mutants is the dampened *PHO* response. For summarizing and illustrating this overall point, we also now include a mega-violin plot, comparing both RNA and RPF (in WT vs. thiolation mutants), for genes related to *PHO*, the PPP, trehalose/gly synthesis, amino acid biosynthesis, nucleotide biosynthesis and ribosomal subunits (Figure 4—figure supplement 3C). This should address any concerns the reviewers might have.

5) Use the proper time frame for the experiment in Figure 2 in order to measure the flux in the PPP and glycolysis pathways in the thiolation mutants, to confirm the rerouting hypothesis.

This is a fair point raised by the reviewers, which requires some clarification. As explained earlier for point 1, this experiment itself was very challenging to carry out, for the following reason: Yeast cells have very high rates of glycolysis and the PPP in standard medium containing glucose as a carbon source. In yeast growing in glucose, because of the very high rate of glycolysis, label saturation is almost instantaneous (in a couple of minutes), immediately after addition of label, for most intermediates in glycolysis and the PPP. Hence, in yeast experiments to measure relative rates of newly synthesized early intermediates of the PPP, using pulse-labels of stable isotopes of glucose (and following labelled carbon incorporation) are challenging. This is due to the fact that it is difficult to label and properly quench metabolites/process samples in time-frames of <5 min, and because most reliable measurements of metabolites require ~10 min post label addition for collection/processing. Indeed, even in more slowly growing, glucose addicted cancer cells, where flux is slower than yeast cells, and where longer time frames of sample collection after stable-isotope addition are used, changes in carbon label incorporation are usually seen with late PPP intermediates, e.g. ribose-5-phosphate. Therefore, even in these studies (performed in timescales of 30min-1 hour post label addition), carbon label incorporation into nucleotides (which come after ribose-5-phosphate) is accepted as an indicator of carbon flux via the PPP and one-carbon metabolism towards nucleotides (e.g., Reid MA et al., Nature Communications2018, 9: 5442, or Lunt et al. Mol Cell 2015). In yeast, given the even faster rates of glycolysis/PPP (compared to typical mammalian cells), it is therefore experimentally reliably possible to process samples only after a slightly longer time interval post label addition, which is why relative label incorporation into nucleotides is preferred. This was the reason we had initially presented data with carbon-label incorporation into NMPs, done ~5-10 minutes after labelled glucose addition. With the new data now included for carbon-label (from glucose) incorporation into ADP and ATP, as well as NMPs in Figures 2B, 2C and related text (also see response to point 3), we fully address this concern.

However, to directly address this point, we optimized even shorter time-points of metabolite extraction after 13C-glucose addition. In ~5 minutes, there already is substantial label saturation, so differences cannot be observed (shown inFigure 2—figure supplement 1B). Therefore, we attempted and reproducibly carried out experiments where metabolites were extracted/processed in <2 minutes after 13C glucose addition. Here, we also observe a significant reduction in relative label incorporation into the late PPP intermediate (and nucleotide precursor) ribose-5-phosphate, in the thiolation mutants (new Figure 2D). This therefore nicely corroborates our conclusions made from the data shown reduced carbon label incorporation into nucleotides (AMP/ADP/ATP) (Figures 2B, C). Experimental note: This difference may be even larger if anyone manages to do such a label addition-quench-process experiment in <1 minute, where label incorporation will be ideal, at <20%. The *only* reference we found where such a flux experiment has been done in yeast in glucose medium, with such a rapid time-frame is (van Heerden et al., Science 2014). We hope all concerns are addressed now with these new data.

6) Discuss the possibility that lower flux in the PPP might result in low NADPH and attendant diminished sulfate assimilation under conditions where sulfur is already limiting. Reduced sulfate assimilation could in turn limit thiolation and amplify the whole regulatory response.

We thank the reviewers for drawing this broader connection to NADPH, and allowing us to include a more nuanced discussion in this regard. We completely agree that nearly all of the thiolation mutant phenotypes can be explained based on NADPH flux and availability, and indeed believe this is so. We had not discussed this in the original submission, because any direct measurement and analysis of NADPH flux and pools in a rigorous, definitive way, is very difficult, and therefore it is difficult to obtain direct data to support this hypothesis. Demonstrated examples of reduction in NADPH flux, coming from decrease flux through the PPP and 1-carbon/folate cycles, have been shown only in rare studies like Fan J et al., Nature 2014, or Li Chen et al. Nature Metabolism (2019). Such studies require a combination of hydrogen-deuterium exchange and possibly mathematical modeling, and is beyond the capability of most labs in the world. Hence, we had cautiously avoided speculation and discussion in this regard. We have now included a nuanced discussion related to this point. As summarized earlier, the thiolation pathway allows a subtler, deeper coupling and integration of various nutrient cues, and modulation of metabolism, for enabling optimal growth. The collective result of the metabolic ‘squeeze’ exerted by the thiolation mutants, by restricting phosphate availability, is a reduction in flux through the PPP and one-carbon cycles, resulting in decreased nucleotide synthesis (as demonstrated). This will also decrease the production of NADPH, and reduce overall reductive biosynthesis capacity, and affect the assimilation of sulfates into sulfur amino acids. Conversely, in the presence of methionine, where there will be maximally thiolated tRNAs, we can connect two distinct observations. In an earlier study, we have shown that the presence of methionine itself increases flux through the PPP and carbon metabolism leading to nucleotide synthesis (Walvekar et al., 2018). We also show this more directly in this study (through measurements of carbon incorporation into nucleotide synthesis in the presence of methionine) (Figure 2G). Here, in the presence of methionine, the loss of thiolation results in a significantly smaller increase in this flux through this arm of carbon metabolism. Hence, the inferred amplified response in relation to tRNA thiolation would be as follows: in the presence of methionine, there is increased PPP and one-carbon flux towards nucleotide synthesis, which is amplified by the maximally thiolated tRNAs by ensuring sufficient phosphate availability. Conversely, reduced sulfate assimilation in turn limits thiolation, and thereby amplify the regulatory response, again using phosphate homeostasis as a control point. This is now an extended paragraph in the Discussion section (third paragraph).

For the information of the reviewers, we include a piece of data consistent with this interpretation: thiolation mutants show hydroxyurea (HU) sensitivity as shown in Figure 3E, and this would be as expected when there are decreased pools of nucleotides, and also explained by decreased NADPH pools available. Here, if this were true, one would predict a growth rescue in the presence of HU if excess reduced glutathione (GSH) is supplemented. This is because oxidized glutathione (GSSG) is reduced to GSH consuming two molecules of NADPH, and hence is a major sink of NADPH consumption. Further, GSH is itself made from sulfur amino acids, and is a major sink of reduced sulfur. By providing excess GSH, this sink can be relieved, and hence free up pools of NADPH. Notably, we now observe a rescue of HU sensitivity in the thiolation mutants upon supplementing GSH (see Author response image 2).

**Author response image 2. respfig2:** 

We however feel that including such data without directly showing changes in NADPH flux is premature. However, we are happy to include a measured Discussion section in this regard, to make a cautious, complete interpretation of our data.

[Editors' note: further revisions were requested prior to acceptance, as described below.]

The manuscript has been improved but there are some remaining issues that need to be addressed before acceptance, as outlined in each of the two reviews shown below. Please make the appropriate revisions and provide a brief point-by-point explanation in response to the remaining reviewers' comments.

Reviewer #1:

I am generally satisfied that the addition of new experimental data and revisions of text address the major concerns with the previous version of the paper, except that the GCN4 mRNA reads still were not added to Figure 4—figure supplement 2.

We apologize for inadvertently not including this earlier. We have now included the GCN4 mRNA reads as an additional panel (Figure 4—figure supplement 2C), and text (subsection “Loss of tRNA thiolation results in reduced phosphate homeostasis (PHO) related transcripts and ribosome-footprints”, third paragraph). As shown, GCN4 mRNA reads are very similar in WT and thiolation mutant cells, and GCN4 transcript amounts do not change in the mutants.

Reviewer #2:

(Partially rephrased by Deputy Editor Detlef Weigel for clarity.)General comment: the essential response points by the authors are difficult to extract from pages of justification which in my opinion turbidify rather than clarify the issues. Several of the original concerns have been satisfactorily addressed, but the following remain:1) Perform new experiments to show that limitation for the sulfur-containing amino acids and/or inorganic sulfate, elicits the same key responses seen for the thiolation mutants, to support the claim that tRNA thiolation is used as a sensor for sulfur levels.

We optimized a simple sulfur-starvation experiment, and performed it as follows: We shifted WT cells growing in standard rich glucose medium, to either minimal glucose medium containing sufficient ammonium sulfate and other trace elements with inorganic sulfur, or to the same medium with sulfur limitation (where chloride salts replace the sulfate salts, as is described in Kankipati et al., 2015). Cells remained in this medium for ~2-4 hours (as described in the Materials and methods). At this time, we collected cells and measured sulfur amino acid related metabolites, and observed substantial decreases in the same, confirming sulfur amino acid limitation (now included as Figure 2—figure supplement 2A). Using this experimental set-up, we compared the key metabolic features of WT cells in this short-term sulfur limitation, with the thiolation mutants as a reference point. The key metabolic features of the thiolation mutants are (i) amino acid accumulation, (ii) decreased nucleotide amounts, and (iii) increased trehalose amounts (due to the re-routing of glucose towards storage carbohydrates). This is also accompanied by an induction of Gcn4 in the thiolation mutants. In WT cells subjected to short-term sulfur limitation, we observe (i) strong accumulation of amino acids (now included as Figure 2—figure supplement 2C), (ii) decreased nucleotides (decreased AMP is shown, now included as Figure 2—figure supplement 2D), and (iii) a strong increase in trehalose synthesis (now included as Figure 2—figure supplement 2). Finally, in a simple sulfur amino acid limitation shift experiment, we also see a strong induction of Gcn4 in the WT cells (now included as Figure 2—figure supplement 2F). For easy comparison, we show the results from the thiolation mutants side-by-side in these figures.

These data compellingly show that sulfur starvation phenocopies tRNA thiolation mutants.

After including these data, we make the following additions to the Results, stating:

“..Furthermore, in a converse experiment, we subjected WT cells to brief inorganic sulfur/sulfur amino acid limitation, to determine the metabolic signature of cells in this condition (experimental design shown in Figure 2—figure supplement 2A).[…] Thus, WT cells subject to sulfur amino acid limitation phenocopied the metabolic signature tRNA thiolation mutants.”

Appropriate additions have also been made to the Materials and methods, and the supplementary figure legends.

We hope these data address all remaining concerns of the reviewer.

Figure 2G shows that absence of methionine elicits a much stronger response than the thiolation mutant does. Indeed, the increased flux toward AMP in the presence of methionine is diminished in the uba4 mutant but stays much higher than in the absence of methionine (same for either WT or uba4) and similar partial effects were observed for GMP (Figure 2—figure supplement 1D). These results indicate that tRNA thiolation contributes (very) partially to the methionine (sulfur) signal (when purine synthesis is used as a readout) suggesting the existence of other (yet unknown) mechanisms connecting sulfur and carbon metabolism. This should be discussed.

We completely agree that the tRNA thiolation does not regulate all methionine dependent responses. In fact we do not suggest these modifications as the sole regulators of methionine dependent responses. For absolute clarity, we have now included the following text stating “Notably, while the extent of this methionine-dependent induction of carbon flux into nucleotides was significantly reduced, it was not completely abolished in the thiolation mutant (Δuba4), indicating the involvement of additional as yet unknown pathways (Figure 2G, and Figure 2—figure supplement 1D).”

4) Expand the analysis of the RNA-Seq data of Figure 5 to show and discuss all of the transcriptional changes for genes in the PPP and trehalose and glycogen pathways.As now shown (Figure 4C and Figure 4—figure supplement 3C) and stated by the authors "there are very minor changes in either transcript or RPF for any of the genes in these pathways". Consequently, the molecular mechanism resulting in rerouting the carbon flux away from the PPP toward trehalose synthesis is clearly not at the RNA or protein expression levels and is not yet elucidated. Hence the last sentence of the Discussion "we biochemically explain how phosphate homeostasis determines flux through different arms of carbon and nitrogen metabolism" appears overstated to me.

We apologize for this overstatement. We have now changed the text, and the final lines read: “Concluding, here we discover that a sulfur amino acid dependent tRNA modification (thiolated U_34_) enables cells to appropriately balance amino acid and nucleotide levels and regulate metabolic state, by controlling phosphate homeostasis. More generally, we suggest how phosphate homeostasis can impact flux through different arms of carbon and nitrogen metabolism.”